# Training Robust Graph Neural Networks by Modeling Noise Dependencies

**Yeonjun In**[1], **Kanghoon Yoon**[1], **Sukwon Yun**[2], **Kibum Kim**[1], **Sungchul Kim**[3]
**Chanyoung Park**[1]*
[1]KAIST    [2]UNC Chapel Hill    [3]Adobe Research
{yeonjun.in, ykhoon08, kb.kim, cy.park}@kaist.ac.kr
swyun@cs.unc.edu
sukim@adobe.com

## Abstract

In real-world applications, node features in graphs often contain noise from various sources, leading to significant performance degradation in GNNs. Although several methods have been developed to enhance robustness, they rely on the unrealistic assumption that noise in node features is independent of the graph structure and node labels, thereby limiting their applicability. To this end, we introduce a more realistic noise scenario, dependency-aware noise on graphs (DANG), where noise in node features create a chain of noise dependencies that propagates to the graph structure and node labels. We propose a novel robust GNN, DA-GNN, which captures the causal relationships among variables in the data generating process (DGP) of DANG using variational inference. In addition, we present new benchmark datasets that simulate DANG in real-world applications, enabling more practical research on robust GNNs. Extensive experiments demonstrate that DA-GNN consistently outperforms existing baselines across various noise scenarios, including both DANG and conventional noise models commonly considered in this field. Our code is available at https://github.com/yeonjun-in/torch-DA-GNN.

## 1  Introduction

In recent years, graph neural networks (GNNs) have demonstrated remarkable achievements in graph representation learning and have been extensively applied in numerous downstream tasks [1, 2, 3, 4]. However, in the majority of real-world scenarios, node features frequently exhibit noise due to various factors, leading to the creation of inaccurate graph representations [5, 6]. For instance, in user-item graphs, users may create fake profiles or posts, and fraudsters and malicious users may write fake reviews or content on items, resulting in noisy node features. Recent studies have revealed the vulnerability of GNNs to such scenarios, highlighting the necessity to design robust GNN models against noisy node features.

To this end, various methods have been proposed to make a huge success in terms of model robustness [5, 6]. These methods are founded on the independent node feature noise (IFN) assumption, which posits that noise in node features does not impact the graph structure or node labels. Under the IFN assumption (Fig. 1(b)), for example, Bob's fake profile does not influence other nodes, which is also explained by the data generating process (DGP) of IFN (See Fig. 2(a)) in which no causal relationships exist among the noisy node features $X$, graph structure $A$, and node labels $Y$.

However, we should rethink: **In real-world applications, can noise in node features truly be isolated from influencing the graph structure or node labels?** Let us explore this through

---

*Corresponding Author

39th Conference on Neural Information Processing Systems (NeurIPS 2025).

examples from social networks (Fig. 1). Consider Bob, who introduces noisy node features by creating fake profiles or posts. Other users, such as Alice and Tom, may then connect with Bob based on his fake profile, resulting in noisy connections that contribute to graph structure noise. Over time, these noises could alter the community associations of Alice and Tom, leading to noisy node labels. Such causal relationships among node features $X$, graph structure $A$, and node label $Y$ (i.e., $A \leftarrow X$, $Y \leftarrow X$, and $Y \leftarrow A$) are depicted in Fig. 2(b).

This scenario underscore an important insight: **In real-world applications, noise in node features may create a chain of noise dependencies that propagate to the graph structure and node labels.** This highlights the pressing need for robust GNNs capable of addressing such noise dependencies, an aspect that has been largely overlooked in current research. Since such noise dependencies are prevalent across a wide range of real-world applications[2] in addition to social networks, failing to address them can result in significant robustness gaps and impede the development of more practical and robust GNN models. However, we observe that existing robust GNN models indeed fail to generalize effectively in such noise scenario since they overlook the underlying relationships among $X$, $A$, and $Y$ within the data generation process.

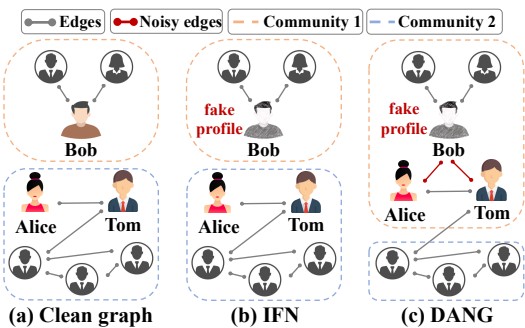

Figure 1: Examples of DANG in social networks: IFN represents independent node feature noise. Under the IFN (b), Bob's noisy features have no effect on the graph structure or node labels. However, in DANG (c), Bob's noisy features can propagate, leading to both structural noise in the graph and label noise.

To enhance the practicality of existing noise assumptions and robust GNNs, we newly introduce a **d**ependency-**a**ware **n**oise on **g**raphs (DANG) and propose a **d**ependency-**a**ware robust **g**raph **n**eural **n**etwork framework (DA-GNN) that directly models the DGP of DANG. We first illustrate the DGP of DANG as shown in Fig. 2(b) (c.f. Sec 3). More precisely, we introduce three observable variables (i.e., $X$, $A$,

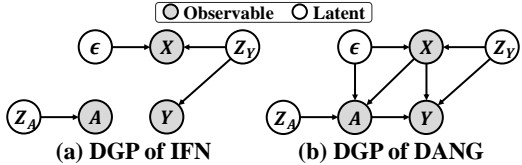

Figure 2: A directed graphical model indicating a DGP of (a) IFN, and (b) DANG.

and $Y$) and three latent variables (i.e., noise incurring variable $\epsilon$, latent clean graph structure $Z_A$, and latent clean node labels $Z_Y$), while defining causal relationships among these variables to represent the data generation process of DANG. We then devise a deep generative model, DA-GNN, that directly captures the causal relationships among the variables in the DGP of DANG by **1)** deriving a tractable learning objective based on variational inference (c.f. Sec 4.1) and **2)** addressing non-trivial technical challenges in implementing the learning objective (c.f. Sec 4.2). Moreover, to rigorously evaluate our proposed method, we propose both synthetic and real-world DANG benchmark datasets. In our experiments, we demonstrate that DA-GNN effectively generalizes not only to DANG but also to other noise assumptions commonly considered in this field of research. This highlights DA-GNN's broader applicability compared to existing robust GNN models. In summary, the main contributions of our paper are as follows:

- We examine the gap between real-world scenarios and the overly simplistic noise assumptions underlying previous robust GNN research, which constrain their practicality.

- To achieve this, we introduce a more realistic noise scenario, DANG, along with a robust model, DA-GNN, improving their applicability in real-world settings.

- DA-GNN addresses DANG by modeling its DGP, resulting in superior robustness in node classification and link prediction tasks under various noise scenarios.

- We propose novel graph benchmark datasets that simulate DANG in real-world applications to evaluate robust GNNs under realistic and plausible noise conditions, thereby promoting practical research in robust graph learning.

---

[2]Additional real-world examples demonstrating the practical existence of such noise are provided in Sec 3.

## 2 Related Work

### 2.1 Noise-Robust GNN

Noise-robust GNNs aim to train robust models under feature, structure, and/or label noise, but most existing approaches focus on only one type of noise.

**Feature noise-robust GNN.** AirGNN [5] identifies and addresses nodes with noisy features based on the hypothesis that they tend to have dissimilar features within their local neighborhoods. Consequently, this approach tackles the noisy node features while assuming that the structure of the input graph is noise-free.

**Structure noise-robust GNN.** RSGNN [7] aims to train a graph structure learner by encouraging the nodes with similar features to be connected. STABLE [8] removes edges with low feature similarity, learns node representations from the modified structure, and constructs a kNN graph as the refined structure. In summary, these methods tackle the noisy graph structure while assuming that node features are noise-free.

**Label noise-robust GNN.** Although there have been many label noise-robust GNNs [9, 10, 11, 12, 13, 14, 15], all these methods are built on the assumption that either node features or graph structures are noise-free. For example, RTGNN [10] uses small-loss approach [16], but nodes with noisy features or structures exhibit large losses, leading to inaccuracies of the approach. TSS [11] mitigates label noise relying on the structural information, which can be noisy.

**Multifaceted noise-robust GNN.** SG-GSR [17] tackles multifaceted structure and feature noise by identifying a clean subgraph within a noisy graph structure and augmenting it using label information. This augmented subgraph serves as supervision for robust graph structure refinement. However, since noisy label information can compromise the augmentation process, SG-GSR relies on the assumption that node labels are free of noise.

In summary, each method assumes the completeness of at least one of the data sources, limiting their practicality.

### 2.2 Generative Approach

[18] devises a generative approach to model the DGP of instance-dependent label noise [19]. However, extending this method to the graph domain introduces significant challenges. It requires handling additional latent variables and complex causal relationships, such as $Z_A$, $\epsilon_A$, $A \leftarrow \epsilon_A$, $A \leftarrow X$, $Y \leftarrow A$, and $A \leftarrow Z_A$, each posing non-trivial obstacles beyond the straightforward extension[3]. WSGNN [20] and GraphGLOW [21] utilize a probabilistic generative approach and variational inference to infer the latent graph structure and node labels. However, they assume noise-free graphs, reducing effectiveness in real-world noisy scenarios.

## 3 Dependency-Aware Noise on Graphs

### 3.1 Formulation

In this section, we define a new graph noise assumption, DANG, and its DGP. In Fig. 2(b), $X$ denotes the node features (potentially noisy), $Y$ denotes the observed node labels (possibly noisy), $A$ denotes the observed edges (which may contain noise), and $\epsilon$ denotes the environment variable causing the noise. $Z_Y$ represents the latent clean node labels, while $Z_A$ does the latent clean graph structure encompassing all potential node connections. We give the explanations for each causal relationship within the DGP of DANG along with the examples in user graphs in social networks:

- $X \leftarrow (\epsilon, Z_Y)$: $\epsilon$ and $Z_Y$ are causes of $X$. For example, users create their profiles and postings (i.e., $X$) regarding their true communities or interests (i.e., $Z_Y$). However, if users decide to display fake profiles for some reason (i.e., $\epsilon$), $\epsilon$ is a cause of the noisy node features $X$.

- $A \leftarrow (Z_A, X)$: $Z_A$ and $X$ are causes of $A$. For instance, the follow relationship among users (i.e., $A$) are made based on their latent relationships (i.e., $Z_A$). However, if a user creates a fake profile (i.e., $X$), some irrelevant users may follow the user based on his/her fake profile, which leads to noisy edges (i.e., $A$).

---

[3]Detailed explanation is outlined in Appendix D.

- $\underline{A \leftarrow \epsilon}$: To provide a broader scope, we also posit that $\epsilon$ is a potential cause of $A$. This extension is well-founded [22, 23], as real-world applications often exhibit graph structure noise originating from various sources in addition to the feature-dependent noise.

- $\underline{Y \leftarrow (Z_Y, X, A)}$: $Z_Y$, $X$, and $A$ are causes of $Y$. To give an example, the true communities (or interests) of users (i.e., $Z_Y$) are leveraged to promote items to targeted users within a community [24]. To detect the communities, both node features and graph structures are utilized. However, if a user has noisy node features (i.e., $X$) or noisy edges (i.e., $A$), the user may be assigned to a wrong community (or interest), which leads to noisy labels (i.e., $Y$).

For simplicity, we assume $\epsilon$ is not a cause of $Y$. This assumption matches real-world scenarios where mislabeling is more likely due to confusing or noisy features rather than arbitrary sources [19]. In other words, label noise in graphs is predominantly caused by confusing or noisy features and graph structure (i.e., $Y \leftarrow (X, A)$), rather than an arbitrary external factor (i.e., $Y \nleftarrow \epsilon$).

### 3.2 Discussion

**1) Under DANG a graph does not contain any noise-free data sources.** This point presents a non-trivial challenge for the existing robust GNN methods to tackle DANG, as they assume the completeness of at least one data source.

**2) DANG is prevalent across diverse domains, including social, e-commerce, web, and biological graphs.** Due to space constraints, detailed statistical evidences and intuitive examples on the existence of DANG in real-world applications are provided in Appendix C.1 and C.2. We acknowledge, however, that not all noise scenarios perfectly align with DANG. For instance, in non-relational domains such as molecular structures or protein–protein interaction networks, the graph structure is fixed and unaffected by node feature noise. Nevertheless, we claim that such cases are rare compared to the broad applicability of DANG across widely studied graph domains, including social, e-commerce, web, and biological (cell-cell) networks.

**3) DANG addresses the practical gap between real-world and the simplistic noise assumptions of previous works.** By introducing the DANG, we examine the practical limitations of existing robust GNN methods and promote further practical advancements in this field.

## 4 Proposed Method: DA-GNN

In this section, we propose a dependency-aware robust GNN framework (DA-GNN) that directly models the DGP of DANG, thereby capturing the causal relationships among the variables that introduce noise. First, we derive the Evidence Lower Bound (ELBO) for the observed data log-likelihood $P(X, A, Y)$ based on the graphical model of DANG (**Sec 4.1**). Subsequently, we introduce a novel deep generative model and training strategy maximizing the derived ELBO to capture the DGP of DANG (**Sec 4.2**).

### 4.1 Problem Formulation

**Notations.** We have an undirected and unweighted graph $\mathcal{G} = \langle \mathcal{V}, \mathcal{E} \rangle$ where $\mathcal{V} = \{v_1, ..., v_N\}$ represents the set of nodes and $\mathcal{E} \in \mathcal{V} \times \mathcal{V}$ indicates the set of edges. Each node $v_i$ has the node features $\mathbf{X}_i \in \mathbb{R}^F$ and node labels $\mathbf{Y}_i \in \{0, 1\}^C$, where $F$ is the number of features for each node and $C$ indicates the number of classes. We represent the observed graph structure using the adjacency matrix $\mathbf{A} \in \mathbb{R}^{N \times N}$, where $\mathbf{A}_{ij} = 1$ if there is an edge connecting nodes $v_i$ and $v_j$, and $\mathbf{A}_{ij} = 0$ otherwise. Throughout this paper, $s(\cdot, \cdot)$ indicates a cosine similarity function and $\rho(\cdot)$ represents the ReLU activation function.

**Tasks: node classification and link prediction.** In the node classification task, we assume the semi-supervised setting where only a portion of nodes are labeled (i.e., $\mathcal{V}^L$). Our objective is to predict the labels of unlabeled nodes (i.e., $\mathcal{V}^U$) by inferring the latent clean node label $Z_Y$. In the link prediction task, our goal is to predict reliable links based on partially observed edges by inferring the latent clean graph structure $Z_A$. It is important to note that, according to the DANG assumption, the observed node features, graph structure, and node labels may contain noise.

**Learning Objective.** We adopt the variational inference framework [25, 18] to optimize the Evidence Lower-BOund (ELBO) of the marginal likelihood for observed data, i.e., $P(X, A, Y)$, rather than

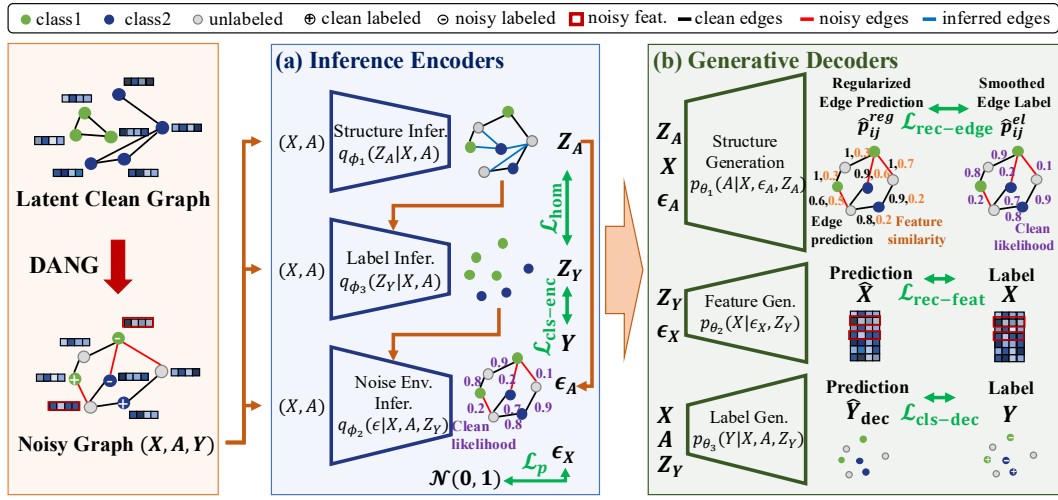

Figure 3: Overall architecture of DA-GNN. (a) With the noisy graph $(X, A, Y)$ as inputs, we design the inference encoders ($\phi_1$, $\phi_2$ and $\phi_3$) and regularizers ($\mathcal{L}_{\text{hom}}$, $\mathcal{L}_{\text{cls-enc}}$, and $\mathcal{L}_{\text{p}}$) to infer $Z_A$, $Z_Y$, $\epsilon_A$, and $\epsilon_X$. (b) Leveraging the inferred latent variables, we formulate the generative decoders ($\theta_1$, $\theta_2$, and $\theta_3$) and reconstruction loss functions ($\mathcal{L}_{\text{rec-edge}}$, $\mathcal{L}_{\text{rec-feat}}$, and $\mathcal{L}_{\text{cls-dec}}$) to capture the causal relationships that generate noise in the graph.

optimizing the marginal likelihood directly. Specifically, we derive the negative ELBO, i.e., $\mathcal{L}_{\text{ELBO}}$, as follows:

$$
\begin{aligned}
\mathcal{L}_{\text{ELBO}} = \\
&- \mathbb{E}_{Z_A \sim q_{\phi_1}(Z_A|X,A)} \mathbb{E}_{\epsilon \sim q_{\phi_2}(\epsilon|X,A,Z_Y)} \left[ \log(p_{\theta_1}(A|X, \epsilon, Z_A)) \right] \\
&- \mathbb{E}_{\epsilon \sim q_{\phi_2}(\epsilon|X,A,Z_Y)} \mathbb{E}_{Z_Y \sim q_{\phi_3}(Z_Y|X,A)} \left[ \log(p_{\theta_2}(X|\epsilon, Z_Y)) \right] \\
&- \mathbb{E}_{Z_Y \sim q_{\phi_3}(Z_Y|X,A)} \left[ \log(p_{\theta_3}(Y|X, A, Z_Y)) \right] \\
&+ kl(q_{\phi_1}(Z_A|X, A) || p(Z_A)) \\
&+ \mathbb{E}_{Z_Y \sim q_{\phi_3}(Z_Y|X,A)} \left[ kl(q_{\phi_2}(\epsilon|X, A, Z_Y) || p(\epsilon)) \right] \\
&+ kl(q_{\phi_3}(Z_Y|X, A) || p(Z_Y))
\end{aligned}
\tag{1}
$$

where $kl(\cdot||\cdot)$ denotes KL divergence. The derivation details are provided in Appendix A. $q_\phi$ indicates inference (encoder) network that approximates the posterior of latent variables, while $p_\theta$ indicates generative (decoder) network that models the likelihood of observed data given latent variables. Our objective is to find the optimal values of network parameters $\phi = \{\phi_1, \phi_2, \phi_3\}$ and $\theta = \{\theta_1, \theta_2, \theta_3\}$ that minimize the value of $\mathcal{L}_{\text{ELBO}}$. By doing so, the encoders and decoders are trained to directly capture the causal relationships among the variables that introduce noise. Consequently, it promotes the accurate inference of the latent clean node label $Z_Y$ and latent clean graph structure $Z_A$ to effectively perform the node classification and link prediction tasks even in the presence of DANG.

## 4.2 Model Instantiations

In this section, we present the details of the practical implementation and optimization of DA-GNN based on the learning objective, $\mathcal{L}_{\text{ELBO}}$. The overall architecture and detailed algorithm of DA-GNN are provided in Fig 3 and Algorithm 1 in Appendix, respectively. The key challenge of the instantiation is how to accurately infer the latent variables $Z_A$, $Z_Y$, and $\epsilon$ in the presence of noisy $X$, $A$, and $Y$. To alleviate the challenge, we design the robust inference encoders (Fig 3(a)) and generative decoders (Fig 3(b)) with the corresponding regularizers (Fig 3(a)) and reconstruction losses (Fig 3(b)). Consequently, the encoders would be able to accurately infer the latent variables by capturing the causal relationships among the variables that introduce noise.

### 4.2.1 Modeling Inference Encoder

In this section, we describe the implementations of the encoders, i.e., $\phi_1$, $\phi_3$, and $\phi_2$, that aim to infer the latent variables, i.e., $Z_A$, $Z_Y$, and $\epsilon$, respectively.

**Modeling $q_{\phi_1}(Z_A|X, A)$.** The objective of modeling $q_{\phi_1}(Z_A|X, A)$ is to accurately infer the latent clean graph structure $Z_A$ that enhances the message passing of a GNN model. We obtain the latent graph $\hat{\mathbf{A}} = \{\hat{p}_{ij}\}_{N \times N}$, where $\hat{p}_{ij} = \rho(s(\mathbf{Z}_i, \mathbf{Z}_j))$ and $\mathbf{Z} = \text{GCN}_{\phi_1}(\mathbf{X}, \mathbf{A})$, and regularize $\hat{\mathbf{A}}$ based on the prior knowledge that pairs of nodes with high $\gamma$-hop subgraph similarity are more likely to form assortative edges [21, 26, 7], thereby encouraging $\hat{\mathbf{A}}$ to predominantly include such edges. This regularization is equivalent to minimizing $kl(q_{\phi_1}(Z_A|X, A)||p(Z_A))$ in Eqn. 1. However, computing $\hat{p}_{ij}$ in every epoch is impractical for large graphs, i.e., $O(N^2)$. To this end, we pre-define a proxy graph based on the subgraph similarity, and compute $\hat{p}_{ij}$ as edge weights on the proxy graph. Please refer to the Appendix B for detailed information on implementation details.

**Modeling $q_{\phi_3}(Z_Y|X, A)$.** The objective of modeling $q_{\phi_3}(Z_Y|X, A)$ is to accurately infer the latent clean node label $Z_Y$. To this end, we instantiate the encoder $\phi_3$ as a GCN classifier. Specifically, we infer $Z_Y$ through $\hat{\mathbf{Y}} = \text{GCN}_{\phi_3}(\mathbf{X}, \hat{\mathbf{A}}) \in \mathbb{R}^{N \times C}$. We introduce the node classification loss $\mathcal{L}_{\text{cls-enc}} = \sum_{i \in \mathcal{V}^L} \text{CE}(\hat{\mathbf{Y}}_i, \mathbf{Y}_i)$, where CE is the cross entropy loss. To further enhance the quality of inference of $Z_Y$, we regularize $Z_Y$ to satisfy class homophily [27] by minimizing the KL divergence between the probability predictions $\hat{\mathbf{Y}}$ of each node and its first order neighbors in $\hat{\mathbf{A}}$. The implemented loss function is given by:

$$\mathcal{L}_{\text{hom}} = \sum_{i \in \mathcal{V}} \frac{\sum_{j \in \mathcal{N}_i} \hat{p}_{ij} \cdot kl(\hat{\mathbf{Y}}_j || \hat{\mathbf{Y}}_i)}{\sum_{j \in \mathcal{N}_i} \hat{p}_{ij}}, \tag{2}$$

where $\mathcal{N}_i$ denotes the set of first-order neighbors of node $v_i$ within $\hat{\mathbf{A}}$. It is worth noting that this regularization is equivalent to minimizing $kl(q_{\phi_3}(Z_Y|X, A)||p(Z_Y))$ in Eqn. 1.

**Modeling $q_{\phi_2}(\epsilon|X, A, Z_Y)$.** To model $q_{\phi_2}(\epsilon|X, A, Z_Y)$, we simplify $q_{\phi_2}(\epsilon|X, A, Z_Y)$ into $q_{\phi_{21}}(\epsilon_X|X, Z_Y)$ and $q_{\phi_{22}}(\epsilon_A|X, A)$, where $\epsilon_X$ and $\epsilon_A$ are independent variables that incur the feature and structure noise, respectively.

The objective of modeling $q_{\phi_{22}}(\epsilon_A|X, A)$ is to accurately infer the structure noise incurring variable $\epsilon_A$ that determines whether each edge is clean or noisy. To this end, we regard $\epsilon_A$ as a set of scores indicating the likelihood of each observed edge being clean or noisy. To estimate the likelihood, we utilize small loss approach [16]. Precisely, we compute the set of link prediction losses as $\{(1 - \hat{p}_{ij}^{el})^2 | (i, j) \in \mathcal{E}\}$, where $\hat{p}_{ij}^{el}$ represents the $\hat{p}_{ij}$ value at the final epoch during early-learning phase. Therefore, an edge with high $\hat{p}_{ij}^{el}$ value can be considered as a clean edge, and we instantiate $\epsilon_A$ as $\{\hat{p}_{ij}^{el} | (i, j) \in \mathcal{E}\}$.

To alleviate the uncertainty of a single training point's loss value, we adopt an exponential moving average (EMA) technique: $\hat{p}_{ij}^{el} \leftarrow \xi \hat{p}_{ij}^{el} + (1 - \xi)\hat{p}_{ij}^{c}$, where $\hat{p}_{ij}^{c}$ indicates the value of $\hat{p}_{ij}$ at the current training point, and $\xi$ indicates the decaying coefficient fixed to 0.9. This approach is equivalent to minimizing $kl(q_{\phi_{22}}(\epsilon_A|X, A)|p(\epsilon_A))$, where $p(\epsilon_A)$ is assumed to follow the same distribution as $q_{\phi_{22}}(\epsilon_A|X, A)$ but with lower variance.

For the encoder $\phi_{21}$, we use an MLP that takes $X$ and $Z_Y$ as inputs and infers $\epsilon_X$. Additionally, we regularize $p(\epsilon_X)$ to follow the standard multivariate normal distribution, which means that a closed form solution of $kl(q_{\phi_{21}}(\epsilon_X|X, Z_Y)||p(\epsilon_X))$ can be obtained as $\mathcal{L}_{\text{p}} = -\frac{1}{2} \sum_{j=1}^{d_2}(1 + \log \sigma_j^2 - \mu_j^2 - \sigma_j^2)$ [28], where $d_2$ is the dimension of a $\epsilon_X$. Note that these two regularization techniques are equivalent to minimizing $\mathbb{E}_{Z_Y \sim q_{\phi_3}}[kl(q_{\phi_2}(\epsilon|X, A, Z_Y)||p(\epsilon))]$ in Eqn. 1.

### 4.2.2 Modeling Generative Decoder

In this section, we describe the implementations of the decoders, i.e., $\theta_1$, $\theta_2$, and $\theta_3$, that generate the observable variables, i.e., $A$, $X$, and $Y$, respectively.

**Modeling $p_{\theta_1}(A|X, \epsilon, Z_A)$.** The probability $p(A|X, \epsilon, Z_A)$ means the likelihood of how well the noisy edge $A$ is reconstructed from the latent graph structure $Z_A$ along with $\epsilon$ and $X$. Hence, we aim to minimize $-\mathbb{E}_{Z_A \sim q_{\phi_1}} \mathbb{E}_{\epsilon \sim q_{\phi_2}}[\log(p_{\theta_1}(A|X, \epsilon, Z_A))]$ to discover the latent graph structure $Z_A$ from which the noisy edge $A$ is reconstructed given noise sources, $X$ and $\epsilon$. We implement it as an edge reconstruction loss forcing the estimated latent structure $\hat{\mathbf{A}}$ to assign greater weights to clean edges and reduce the influence of noisy edges, which is defined as $\mathcal{L}_{\text{rec-edge}}$: which is defined as $\mathcal{L}_{\text{rec-edge}}$:

$$\mathcal{L}_{\text{rec-edge}} = \frac{N}{|\mathcal{E}| + |\mathcal{E}^-|} \left( \sum_{(i,j) \in \mathcal{E}} (\hat{p}_{ij}^{reg} - \hat{p}_{ij}^{el})^2 + \sum_{(i,j) \in \mathcal{E}^-} (\hat{p}_{ij} - 0)^2 \right), \quad (3)$$

where $\mathcal{E}$ and $\mathcal{E}^-$ denote the positive edges and randomly sampled negative edges, respectively. To compute $\mathcal{L}_{\text{rec-edge}}$, we employ regularizations on both the predictions (i.e., $\hat{p}_{ij}^{reg}$) and labels (i.e., $\hat{p}_{ij}^{el}$) since the observed graph structure $A$ contains noisy edges incurred by $X$ and $\epsilon$, which introduce inaccurate supervision.

More precisely, the regularized prediction $\hat{p}_{ij}^{reg}$ is defined as: $\hat{p}_{ij}^{reg} = \theta_1 \hat{p}_{ij} + (1 - \theta_1) s(\mathbf{X}_i, \mathbf{X}_j)$. The main idea is to penalize $\hat{p}_{ij}$ when $s(\mathbf{X}_i, \mathbf{X}_j)$ is high, as the edge between $v_i$ and $v_j$ is potentially noisy due to the influence of noisy $X$. To regularize labels, we adopt label smoothing approach by $\hat{p}_{ij}^{el} \in [0.9, 1]$, enhancing the robustness in the presence of noisy supervision. When an edge is regarded as noisy (i.e., with a low $\hat{p}_{ij}^{el}$), its label is close to 0.9[4], while an edge considered clean (i.e., with a high $\hat{p}_{ij}^{el}$) has a label close to 1.

**Modeling $p_{\theta_2}(X|\epsilon, Z_Y)$).** The term $p(X|\epsilon, Z_Y)$ indicates how well the noisy node feature $X$ is reconstructed from the latent clean label $Z_Y$ along with $\epsilon$. Hence, we aim to minimize $-\mathbb{E}_{\epsilon \sim q_{\phi_2}} \mathbb{E}_{Z_Y \sim q_{\phi_3}} [\log(p_{\theta_2}(X|\epsilon, Z_Y))]$. To do so, the decoder needs to rely on the information contained in $Z_Y$, which essentially encourages the value of $Z_Y$ to be meaningful for the prediction process, i.e., generating $X$. It is implemented as a feature reconstruction loss $\mathcal{L}_{\text{rec-feat}}$, where the decoder $\theta_2$ is composed of an MLP that takes $\epsilon_X$ and $Z_Y$ as inputs and reconstructs node features. Note that the reparametrization trick [28] is used for sampling $\epsilon_X$ that follows the standard normal distribution.

**Modeling $p_{\theta_3}(Y|X, A, Z_Y)$.** The term $p(Y|X, A, Z_Y)$ means the transition relationship from the latent clean label $Z_Y$ to the noisy label $Y$ of an instance, i.e., how the label noise was generated [18]. For this reason, maximizing $\log(p_{\theta_3}(Y|X, A, Z_Y))$ would let us discover the latent true label $Z_Y$ from which the noisy label $Y$ is generated given an instance, i.e., $X$ and $A$. Hence, we aim to maximize the log likelihood, which is implemented as minimizing a node classification loss $\mathcal{L}_{\text{cls-dec}}$. Specifically, the decoder $\theta_3$ is composed of a GCN classifier: $\hat{\mathbf{Y}}_{\text{dec}} = \text{GCN}_{\theta_3}(\mathbf{X}, \mathbf{A}, \hat{\mathbf{Y}}) \in \mathbb{R}^{N \times C}$. Note that such a learning objective is equivalent to minimizing $-\mathbb{E}_{Z_Y \sim q_{\phi_3}} [\log(p_{\theta_3}(Y|X, A, Z_Y))]$ in Eqn. 1.

### 4.2.3 Model Training

The overall learning objective can be written as follows and DA-GNN is trained to minimize $\mathcal{L}_{\text{final}}$:

$$\mathcal{L}_{\text{final}} = \mathcal{L}_{\text{cls-enc}} + \lambda_1 \mathcal{L}_{\text{rec-edge}} + \lambda_2 \mathcal{L}_{\text{hom}} + \lambda_3 (\mathcal{L}_{\text{rec-feat}} + \mathcal{L}_{\text{cls-dec}} + \mathcal{L}_{\text{p}}), \quad (4)$$

where $\lambda_1$ and $\lambda_2$ are the balancing coefficients. $\lambda_3$ is fixed to 0.001. In our pilot experiments, $\mathcal{L}_{\text{rec-feat}}$, $\mathcal{L}_{\text{cls-dec}}$, and $\mathcal{L}_{\text{p}}$ terms have a relatively minor impact on the model's performance compared to the others. As a result, we have made a strategic decision to simplify the hyperparameter search process and improve the practicality of DA-GNN by sharing the coefficient $\lambda_3$ among these three loss terms.

## 5 Experiments

**Datasets.** We evaluate DA-GNN and baselines on *five commonly used benchmark datasets* and *two newly introduced datasets*, Auto and Garden, which are generated upon Amazon review data [30, 31] to mimic DANG on e-commerce systems (Refer to Appendix E.2.2 for details). The details of the datasets are given in Appendix E.1.

**Experimental Details.** We evaluated DA-GNN in both node classification and link prediction tasks, comparing it with noise-robust GNNs and generative GNN methods. For a thorough evaluation, we create synthetic and real-world DANG benchmark datasets, with details in Appendix E.2. We also account for other noise scenarios, commonly considered in this research field, following [8, 5, 10]. Further details about the baselines, evaluation protocol, and implementation details can be found in Appendix E.3, E.4, and E.5, respectively.

---

[4]The value 0.9 is selected following [29].

Table 1: Node classification accuracy (%) under synthetic DANG. OOM indicates out of memory on 24GB RTX3090.

| Dataset | Setting | WSGNN | GraphGLOW | AirGNN | ProGNN | RSGNN | STABLE | EvenNet | NRGNN | RTGNN | SG-GSR | DA-GNN |
|---|---|---|---|---|---|---|---|---|---|---|---|---|
| Cora | Clean | **86.2±0.1** | 85.2±0.7 | 85.0±0.2 | 85.3±0.4 | **86.2±0.5** | 86.1±0.2 | **86.2±0.0** | **86.2±0.2** | 86.1±0.2 | 85.7±0.1 | 86.2±0.7 |
| | DANG-10% | 80.7±0.3 | 79.7±0.2 | 79.7±0.5 | 79.6±0.7 | 81.9±0.3 | 82.2±0.7 | 80.7±0.7 | 81.0±0.5 | 81.8±0.3 | 82.7±0.1 | **82.9±0.6** |
| | DANG-30% | 70.0±0.6 | 71.6±0.5 | 71.5±0.8 | 74.5±0.1 | 71.9±0.5 | 74.3±0.3 | 65.2±1.7 | 73.5±0.8 | 72.6±1.5 | 76.1±0.2 | **78.2±0.3** |
| | DANG-50% | 55.9±1.1 | 59.6±0.1 | 56.2±0.8 | 66.4±0.4 | 59.9±0.5 | 62.8±2.4 | 47.1±1.8 | 61.9±1.4 | 60.9±0.4 | 64.3±0.5 | **69.7±0.6** |
| Citeseer | Clean | 76.6±0.6 | 76.5±1.0 | 71.5±0.2 | 72.6±0.5 | 75.8±0.4 | 74.6±0.6 | 76.4±0.5 | 75.0±1.3 | 76.1±0.4 | 75.3±0.3 | **77.3±0.6** |
| | DANG-10% | 72.8±0.8 | 71.4±0.8 | 66.2±0.7 | 67.5±0.6 | 73.3±0.5 | 71.5±0.3 | 71.1±0.4 | 71.9±0.3 | 73.2±0.2 | 74.2±0.5 | **74.3±0.9** |
| | DANG-30% | 63.3±0.7 | 60.6±0.2 | 58.0±0.4 | 61.0±0.2 | 63.9±0.5 | 62.5±1.4 | 61.2±0.6 | 62.5±0.7 | 64.2±1.9 | **65.6±1.0** | 65.6±0.6 |
| | DANG-50% | 53.4±0.6 | 48.8±0.6 | 50.0±0.6 | 53.3±0.2 | 55.3±0.4 | 54.7±1.7 | 47.2±1.1 | 52.6±0.9 | 54.2±1.8 | 54.8±1.8 | **59.0±1.8** |
| Photo | Clean | 92.9±0.3 | 94.2±0.4 | 93.5±0.1 | 90.1±0.2 | 93.6±0.8 | 93.4±0.1 | 94.5±0.4 | 90.3±1.7 | 91.3±0.6 | 94.3±0.1 | **94.8±0.3** |
| | DANG-10% | 83.9±1.8 | 92.1±0.8 | 87.3±0.9 | 84.3±0.1 | 92.1±0.2 | 92.2±0.1 | 92.6±0.0 | 84.3±1.3 | 89.4±0.5 | 93.0±0.1 | **93.2±0.2** |
| | DANG-30% | 51.9±6.8 | 88.4±0.2 | 67.8±4.3 | 74.7±0.2 | 86.6±1.0 | 88.0±1.0 | 89.6±0.2 | 69.0±2.2 | 86.4±0.5 | 89.3±0.3 | **90.5±0.4** |
| | DANG-50% | 31.9±5.6 | 85.4±0.6 | 57.8±0.7 | 48.9±0.5 | 75.6±2.6 | 80.2±1.8 | 84.6±0.4 | 57.5±1.8 | 79.2±0.3 | 84.1±0.4 | **87.6±0.2** |
| Comp | Clean | 83.1±3.1 | 91.3±0.9 | 83.4±1.2 | 83.9±0.8 | 91.1±0.1 | 90.2±0.2 | 87.5±1.0 | 87.3±1.0 | 87.3±1.0 | 91.3±0.7 | **92.2±0.0** |
| | DANG-10% | 75.0±1.2 | 88.0±0.7 | 76.8±1.8 | 72.0±0.2 | 88.1±0.7 | 85.9±0.5 | 87.6±0.7 | 85.7±0.9 | 85.9±0.1 | 89.5±0.5 | **89.8±0.2** |
| | DANG-30% | 48.5±5.8 | 84.9±0.4 | 59.2±0.9 | 66.9±0.8 | 81.7±0.2 | 80.4±1.0 | 84.8±0.5 | 74.8±3.5 | 77.0±1.5 | 84.5±0.4 | **86.9±0.3** |
| | DANG-50% | 39.6±4.0 | 80.1±0.5 | 44.1±1.4 | 43.3±0.3 | 73.9±2.3 | 68.8±1.3 | 77.5±1.9 | 65.3±3.2 | 69.4±0.3 | 78.6±0.6 | **82.2±0.4** |
| Arxiv | Clean | OOM | OOM | 58.0±0.4 | OOM | OOM | OOM | 65.7±0.6 | OOM | 60.4±0.5 | OOM | **67.4±0.4** |
| | DANG-10% | OOM | OOM | 50.6±0.5 | OOM | OOM | OOM | 58.4±1.2 | OOM | 54.3±0.4 | OOM | **59.7±0.8** |
| | DANG-30% | OOM | OOM | 36.8±0.3 | OOM | OOM | OOM | 47.4±2.5 | OOM | 45.0±0.6 | OOM | **49.9±0.5** |
| | DANG-50% | OOM | OOM | 26.1±0.2 | OOM | OOM | OOM | 38.0±4.1 | OOM | 38.4±0.8 | OOM | **44.0±1.2** |

Table 2: Node classification (NC) and link prediction (LP) under real-world DANG (Accuracy for NC and ROC-AUC for LP).

| Task | Dataset | Setting | WSGNN | GraphGLOW | AirGNN | ProGNN | RSGNN | STABLE | EvenNet | NRGNN | RTGNN | SG-GSR | DA-GNN |
|---|---|---|---|---|---|---|---|---|---|---|---|---|---|
| NC | Auto | Clean | 71.8±4.3 | 77.9±1.2 | 69.5±0.8 | 63.2±0.2 | 69.5±0.4 | 71.6±0.9 | 73.4±0.5 | 74.3±0.8 | 76.4±0.2 | 78.3±0.3 | **79.3±0.2** |
| | | + DANG | 57.7±1.3 | 59.4±0.8 | 53.9±0.1 | 48.6±0.3 | 56.8±0.9 | 57.5±0.2 | 57.1±2.1 | 55.8±1.0 | 59.6±0.8 | **62.0±1.1** | 61.4±0.4 |
| | Garden | Clean | 87.4±0.2 | 88.5±0.9 | 78.3±1.5 | 78.7±0.1 | 83.3±1.2 | 84.2±0.5 | 85.7±0.5 | 87.7±0.4 | 87.8±0.2 | 88.1±0.3 | **88.7±0.3** |
| | | + DANG | 77.6±0.8 | 78.1±1.5 | 66.1±1.7 | 73.0±0.4 | 76.2±0.5 | 77.2±3.3 | 75.6±2.4 | 76.1±0.2 | 76.0±0.6 | 80.2±0.4 | **80.2±0.8** |
| LP | Auto | Clean | 81.8±0.1 | 86.2±0.3 | 60.2±0.2 | 74.8±0.3 | 87.2±0.8 | 78.6±0.1 | 86.8±0.1 | 76.6±1.3 | 84.4±0.1 | 82.2±8.3 | **88.2±0.3** |
| | | + DANG | 69.1±0.6 | **74.8±0.2** | 57.9±0.4 | 56.7±0.5 | 65.0±0.2 | 57.3±0.1 | 70.5±0.2 | 47.5±1.7 | 72.2±0.2 | 65.6±7.4 | 73.6±0.6 |
| | Garden | Clean | 84.7±0.2 | 90.2±0.5 | 62.0±0.1 | 83.5±0.6 | 91.2±0.4 | 85.2±0.2 | 89.2±0.3 | 87.0±0.9 | 90.4±0.3 | 89.2±3.8 | **92.6±0.2** |
| | | + DANG | 84.6±0.7 | 90.1±0.4 | 58.2±0.5 | 83.3±0.5 | 91.2±0.5 | 85.0±0.1 | 90.0±0.7 | 58.6±4.5 | 90.4±0.2 | 86.0±7.2 | **92.4±0.4** |

## 5.1 Main Results

**1) DA-GNN demonstrates superior robustness compared to baseline methods in handling noise dependencies represented by DANG.** We first evaluate DA-GNN under synthetic DANG datasets. Table 1 shows that DA-GNN consistently outperforms all baselines in DANG scenarios, especially when noise levels are high. This superiority is attributed to the fact that DA-GNN captures the causal relationships involved in the DGP of DANG, while the baselines overlook such relationships, leading to their model designs assuming the completeness of at least one data source. Moreover, we investigate the robustness under our proposed real-world DANG datasets, Auto and Garden, that we simulate noise dependencies within e-commerce systems. In Table 2, we observe that DA-GNN outperforms the baselines under real-world DANG on both the node classification and link prediction tasks. This indicates that DA-GNN works well not only under artificially generated noise, but also under noise scenarios that are plausible in real-world applications.

**2) DA-GNN also shows comparable or better performance than baselines under other noise scenarios, commonly considered in this research field.** Specifically, we evaluate the robustness of DA-GNN under commonly utilized node feature noise [5], structure noise [8], and node label noise scenarios [9] on Cora dataset[5]. In Fig 4, we observe DA-GNN shows consistent superiority or competitive performance compared to existing robust GNNs.

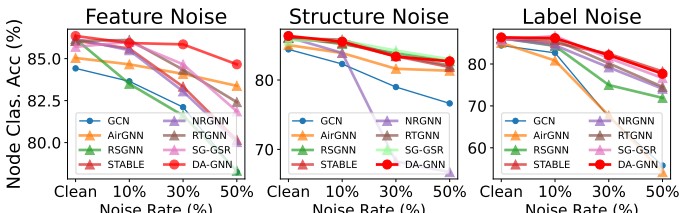

Figure 4: Node classification under node feature noise, structure noise, and node label noise scenarios, which are commonly considered in robust GNN research field, on Cora dataset.

We attribute the robustness of DA-GNN under the noise in node features to the graph structure learning module that accurately infers the latent graph structure $Z_A$. The utilization of abundant local neighborhoods acquired through the inference

---

[5]Additional results on other datasets are outlined in Fig 10, 11, and 12 in Appendix.

of $Z_A$ enables effective smoothing for nodes with noisy features, leveraging the information within these neighborhoods. We attribute the effectiveness of DA-GNN under the noise in graph structures to inferring the robust latent clean graph structure. In other words, the inference of the latent clean graph structure $Z_A$ assigns greater weights to latent clean edges and lower weights to observed noisy edges by employing regularizations on both the edge predictions and labels, thereby mitigating structural noise. For the noise in node labels, we argue that the effectiveness of DA-GNN stems from the accurate inference of the latent clean structure. Specifically, the inferred latent node label $Z_Y$ is regularized using the inferred latent structure $Z_A$ to meet the homophily assumption (i.e., $\mathcal{L}_{\text{hom}}$). Leveraging the clean neighbor structure, this regularization technique has been demonstrated to effectively address noisy labels [32].

**3) DA-GNN outperforms all baselines under an extreme noise scenario.** In addition to a single type of noise, we explore a more challenging noise scenario where all three types of noises occur simultaneously, denoted as extreme noise. It is important to note that each type of noise does not affect the occurrence of the other types of noise, in contrast to DANG. In Fig 5, DA-GNN consistently outperforms the robust GNNs under extreme noise.

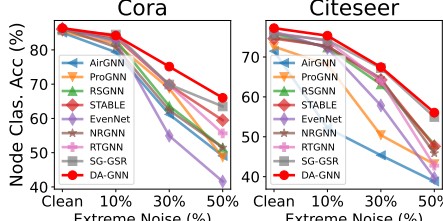

Figure 5: Node classification under extreme noise scenario.

**SUMMARY: DA-GNN has a broader range of applicability than the existing robust GNNs under various noise scenarios.** Based on the above results, we assert that modeling the DGP of DANG offers significant advantages for robustness, both under DANG and independently occurring feature, structure, or label noise, as DA-GNN is inherently capable of handling each type of noise. In contrast, the baseline methods assume the completeness of at least one of the data sources, resulting in a significant performance drop when the noise rate is high.

## 5.2 Ablation Studies on DA-GNN

To emphasize the importance of directly capturing the causal relationships among variables in the DGP of DANG, i.e., $Y \leftarrow (X, A)$, $A \leftarrow X$, and $A \leftarrow \epsilon$, we remove them one by one from the graphical model of DANG (See Fig 2(b), and then design deep generative models based on the DGPs in a similar manner to DA-GNN. The graphical models of the derived DGPs are illustrated in Fig 6. In Table 3, we observe that as more causal relationships are removed from the DGP of DANG, the node classification performance decreases. Below, we offer explanations for this observation from the perspective of model derivation.

Table 3: Ablation studies of various DGPs from Fig 6. Case 3 removes $Y \leftarrow (X, A)$; Case 2 additionally removes $A \leftarrow X$; Case 1 additionally removes $A \leftarrow \epsilon$, equivalent to IFN (Fig 2).

| Dataset | Setting | (a) Case 1 | (b) Case 2 | (c) Case 3 | **Proposed** |
|---|---|---|---|---|---|
| Cora | Clean | 84.6±0.4 | 84.8±0.4 | **86.2±0.2** | **86.2±0.7** |
| | DANG-10% | 77.4±0.3 | 77.3±0.3 | **83.2±0.3** | 82.9±0.6 |
| | DANG-30% | 68.3±0.4 | 68.5±0.2 | 77.3±0.4 | **78.2±0.3** |
| | DANG-50% | 55.2±0.2 | 56.1±0.3 | 68.7±0.3 | **69.7±0.6** |
| Citeseer | Clean | 76.7±0.9 | 76.8±0.8 | 76.5±0.9 | **77.3±0.6** |
| | DANG-10% | 69.5±0.3 | 69.5±0.4 | 73.2±0.1 | **74.3±0.9** |
| | DANG-30% | 57.2±1.1 | 57.7±0.5 | 65.5±0.7 | **65.6±0.6** |
| | DANG-50% | 49.2±0.5 | 48.7±0.2 | 57.6±2.5 | **59.0±1.8** |

Figure 6: Graphical models of DGPs derived from DANG.

**1)** Removing $Y \leftarrow (X, A)$, i.e., Fig 6(c), simplifies $-\mathbb{E}_{Z_Y \sim q_{\phi_3}}[\log(p_{\theta_3}(Y|X, A, Z_Y))]$ to $-\mathbb{E}_{Z_Y \sim q_{\phi_3}}[\log(p_{\theta_3}(Y|Z_Y))]$. This simplification hinders the accurate modeling of the label transition relationship from $Z_Y$ to the noisy label $Y$, resulting in a degradation of model performance under DANG.

**2)** Additionally, when excluding $A \leftarrow X$, i.e., Fig 6(b), the inference of $Z_A$ and $Z_Y$ is simplified as follows: $q_{\phi_1}(Z_A|X, A)$ to $q_{\phi_1}(Z_A|A)$ and $q_{\phi_3}(Z_Y|X, A)$ to $q_{\phi_3}(Z_Y|X)$. Furthermore, the loss term $-\mathbb{E}_{Z_A \sim q_{\phi_1}}\mathbb{E}_{\epsilon \sim q_{\phi_2}}[\log(p_{\theta_1}(A|X, \epsilon, Z_A))]$ is also simplified to $-\mathbb{E}_{Z_A \sim q_{\phi_1}}\mathbb{E}_{\epsilon \sim q_{\phi_2}}[\log(p_{\theta_1}(A|\epsilon, Z_A))]$. These simplifications significantly hinder the accurate inference of $Z_A$ and $Z_Y$, resulting in a notable performance degradation.

**3)** Eliminating $A \leftarrow \epsilon$, as in Fig 6(a), simplifies $-\mathbb{E}_{Z_A \sim q_{\phi_1}}\mathbb{E}_{\epsilon \sim q_{\phi_2}}[\log(p_{\theta_1}(A|\epsilon, Z_A))]$ to $-\mathbb{E}_{Z_A \sim q_{\phi_1}}\mathbb{E}_{\epsilon \sim q_{\phi_2}}[\log(p_{\theta_1}(A|Z_A))]$. This simplification hinders the robustness of the inferred $Z_A$, since the simplified loss excludes label regularization from the model training process, ultimately resulting in performance degradation.

### 5.3 Complexity Analysis on DA-GNN

We provide both theoretical and empirical complexity analyses of training DA-GNN. Our findings show that DA-GNN achieves superior performance compared to baseline methods while maintaining acceptable training times. For a detailed discussion and comprehensive results, refer to Appendix F.1.

### 5.4 Sensitivity Analysis

We analyze the sensitivity of our proposed method DA-GNN in terms of its hyperparameters $\lambda_1$, $\lambda_2$, $k$, $\theta$, and $\gamma$. Our observations indicate that DA-GNN consistently exhibit best performance regardless of their values. Among these, $k$ plays a critical role and requires some tuning. But, as the search space is relatively small, we consider this acceptable. For a more comprehensive discussion and detailed results, please see Appendix F.2.

### 5.5 Robustness Evaluation under Variants of DANG

We analyze the robustness of DA-GNN across different variants of DANG by varying the hyperparameter settings used in dataset generation. Specifically, in the generation process of our synthetic DANG, we have three variables: 1) the overall noise rate, 2) the amount of noise dependency ($X \rightarrow A$, ($X \rightarrow Y$, ($A \rightarrow Y$), and 3) the amount of independent structure noise ($\epsilon \rightarrow A$). For the generation process of our real-world DANG, we have 1) the number of fraudsters (i.e., nodes with noisy features) and 2) the activeness of fraudsters (i.e., the amount of structure noise they introduce). As a result, label noise also increases accordingly, in proportion to the amount of generated feature and structure noise.

Detailed results in Appendix F.3 show that DA-GNN consistently outperforms all baselines across varying levels of both synthetic and real-world DANG, underscoring its robustness and practical applicability under diverse noise conditions.

### 5.6 Qualitative Analysis on DA-GNN

We conduct qualitative analyses to verify how well DA-GNN infers the latent variables $\epsilon_A$ and $Z_A$. For a detailed setting and results, please refer to Appendix F.4.

## 6 Conclusion

This study investigates the practical gap between real-world scenarios and the simplistic noise assumptions in terms of node features underlying previous robust GNN research. To bridge this gap, we newly introduce a more realistic graph noise scenario called dependency-aware noise on graphs (DANG), and present a deep generative model, DA-GNN, that effectively captures the causal relationships among variables in the DGP of DANG. We also propose novel graph benchmarks that simulate DANG within real-world applications, which fosters practical research in this field. We demonstrate DA-GNN has a broader applicability than the existing robust GNNs under various noise scenarios.

## 7 Limitations and Future Works

Despite broader applicability of the DANG and DA-GNN, they do not perfectly cover all possible noise scenarios. One direction to enhance their practicality is to incorporate $X \leftarrow A$, suggesting graph structure noise can inevitably lead to node feature noise. By doing so, a broader range of noise scenarios could be addressed, further improving practical applicability. A detailed discussion on this topic is provided in Appendix C.3.

## Acknowledgements

This work was supported by the Institute of Information & Communications Technology Planning & Evaluation(IITP) grant funded by the Korea government(MSIT) (RS-2025-02304967), IITP grant funded by the Korea government(MSIT) (RS-2022-II220157), National Research Foundation of Korea(NRF) funded by Ministry of Science and ICT (RS-2022-NR068758).

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

# A Derivation Details of Evidence Lower BOund (ELBO)

We commence by modeling joint distribution $P(X, A, Y)$. We assume that the joint distribution $P(X, A, Y)$ is differentiable nearly everywhere regarding both $\theta$ and the latent variables $(\epsilon, Z_A, Z_Y)$. Note that the generative parameter $\theta$ serves as the decoder network that models the distribution $P(X, A, Y)$. The joint distribution of $P(X, A, Y)$ can be represented as:

$$p_\theta(X, A, Y) = \int_\epsilon \int_{Z_A} \int_{Z_Y} p_\theta(X, A, Y, \epsilon, Z_A, Z_Y) d\epsilon dZ_A dZ_Y. \tag{5}$$

However, computing this evidence integral is either intractable to calculate in closed form or requires exponential time. As the evidence integral is intractable for computation, calculating the conditional distribution of latent variables $p_\theta(\epsilon, Z_A, Z_Y | X, A, Y)$ is also intractable:

$$p_\theta(\epsilon, Z_A, Z_Y | X, A, Y) = \frac{p_\theta(X, A, Y, \epsilon, Z_A, Z_Y)}{p_\theta(X, A, Y)}. \tag{6}$$

To infer the latent variables, we introduce an inference network $\phi$ to model the variational distribution $q_\phi(\epsilon, Z_A, Z_Y | X, A, Y)$, which serves as an approximation to the posterior $p_\theta(\epsilon, Z_A, Z_Y | X, A, Y)$. To put it more concretely, the posterior distribution can be decomposed into three distributions determined by trainable parameters $\phi_1$, $\phi_2$, and $\phi_3$. Based on the observed conditional independence relationships [6], we decompose $q_\phi(\epsilon, Z_A, Z_Y | X, A, Y)$ as follows:

$$q_\phi(\epsilon, Z_A, Z_Y | X, A, Y) = q_{\phi_1}(Z_A | X, A, \epsilon) q_{\phi_2}(\epsilon | X, A, Z_Y) q_{\phi_3}(Z_Y | X, A, Y). \tag{7}$$

For simplicity, we introduce two additional assumptions. First, when the node features $X$ and observed graph structure $A$ are given, latent clean graph structure $Z_A$ is conditionally independent from the noise-incurring variable $\epsilon$, i.e., $q_{\phi_1}(Z_A | X, A, \epsilon) = q_{\phi_1}(Z_A | X, A)$. Second, when $X$ and $A$ are given, latent clean labels $Z_Y$ is conditionally independent from the observed node labels $Y$, i.e., $q_{\phi_3}(Z_Y | X, A, Y) = q_{\phi_3}(Z_Y | X, A)$. This approximation, known as the mean-field method, is a prevalent technique utilized in variational inference-based methods [33, 20]. As a result, we can simplify Eqn. 7 as follows:

$$q_\phi(\epsilon, Z_A, Z_Y | X, A, Y) = q_{\phi_1}(Z_A | X, A) q_{\phi_2}(\epsilon | X, A, Z_Y) q_{\phi_3}(Z_Y | X, A). \tag{8}$$

To jointly optimize the parameter $\phi$ and $\theta$, we adopt the variational inference framework [25, 18] to optimize the Evidence Lower-BOund (ELBO) of the marginal likelihood for observed data, rather than optimizing the marginal likelihood directly. Specifically, we derive the ELBO for the observed data log-likelihood $P(X, A, Y)$. First, we factorize the joint distribution $P(X, A, Y, \epsilon, Z_A, Z_Y)$ based on the graphical model in Fig. 2(b) in the main paper:

$$
\begin{aligned}
&P(X, A, Y, \epsilon, Z_A, Z_Y) \\
&= P(\epsilon)P(Z_A)P(Z_Y)P(X|\epsilon, Z_Y)P(A|\epsilon, X, Z_A)P(Y|X, A, Z_Y).
\end{aligned} \tag{9}
$$

Thus, the conditional distribution $P_\theta(X, A, Y | \epsilon, Z_A, Z_Y)$ can be represented as follows:

$$P_\theta(X, A, Y | \epsilon, Z_A, Z_Y) = P_{\theta_1}(X|\epsilon, Z_Y)P_{\theta_2}(A|\epsilon, X, Z_A)P_{\theta_3}(Y|X, A, Z_Y). \tag{10}$$

Recall that the conditional distribution $q_\phi(\epsilon, Z_A, Z_Y | X, A, Y)$ is factorized as in Eqn. 8. Now, we derive the ELBO for the observed data log-likelihood $P(X, A, Y)$:

---

[6]We observe the following conditional independence relationships in Fig. 2(b): (1) $Z_A \perp Y | X, A, \epsilon$, (2) $Z_A \perp Z_Y | A, X, \epsilon$, (3) $\epsilon \perp Y | Z_Y, X, A$.

$$\log p_\theta(X, A, Y) = \log \int_\epsilon \int_{Z_A} \int_{Z_Y} p_\theta(X, A, Y, \epsilon, Z_A, Z_Y) d\epsilon dZ_A dZ_Y$$

$$= \log \int_\epsilon \int_{Z_A} \int_{Z_Y} p_\theta(X, A, Y, \epsilon, Z_A, Z_Y) \frac{q_\phi(\epsilon, Z_A, Z_Y | X, A, Y)}{q_\phi(\epsilon, Z_A, Z_Y | X, A, Y)}$$

$$= \log \mathbb{E}_{(\epsilon, Z_A, Z_Y) \sim q_\phi(\epsilon, Z_A, Z_Y | X, A, Y)} \left[ \frac{p_\theta(X, A, Y, \epsilon, Z_A, Z_Y)}{q_\phi(\epsilon, Z_A, Z_Y | X, A, Y)} \right]$$

$$\geq \mathbb{E}_{(\epsilon, Z_A, Z_Y) \sim q_\phi(\epsilon, Z_A, Z_Y | X, A, Y)} \left[ \log \frac{p_\theta(X, A, Y, \epsilon, Z_A, Z_Y)}{q_\phi(\epsilon, Z_A, Z_Y | X, A, Y)} \right] := \mathrm{ELBO}$$

$$= \mathbb{E}_{(\epsilon, Z_A, Z_Y) \sim q_\phi(\epsilon, Z_A, Z_Y | X, A, Y)} \left[ \log \frac{p(\epsilon) p(Z_A) p(Z_Y)}{q_\phi(\epsilon, Z_A, Z_Y | X, A, Y)} \right.$$

$$\left. + \log \frac{p_{\theta_1}(A | X, \epsilon, Z_A) p_{\theta_2}(X | \epsilon, Z_Y) p_{\theta_3}(Y | X, A, Z_Y)}{q_\phi(\epsilon, Z_A, Z_Y | X, A, Y)} \right]$$

$$= \mathbb{E}_{(\epsilon, Z_A, Z_Y) \sim q_\phi(\epsilon, Z_A, Z_Y | X, A, Y)} \left[ \log(p_{\theta_1}(A | X, \epsilon, Z_A)) \right.$$

$$\left. + \log(p_{\theta_2}(X | \epsilon, Z_Y)) + \log(p_{\theta_3}(Y | X, A, Z_Y)) \right]$$

$$+ \mathbb{E}_{(\epsilon, Z_A, Z_Y) \sim q_\phi(\epsilon, Z_A, Z_Y | X, Y, A)} \left[ \log \frac{p(\epsilon) p(Z_A) p(Z_Y)}{q_\phi(\epsilon, Z_A, Z_Y | X, A, Y)} \right] \tag{11}$$

The last equation of Eq. 11 can be more simplified. We present the simplified results in Eqn. 12, 13, 14, and 15, where we abuse the notation $\mathbb{E}_{Z_A \sim q_{\phi_1}(Z_A | X, A)}$, $\mathbb{E}_{\epsilon \sim q_{\phi_2}(\epsilon | X, A, Z_Y)}$, and $\mathbb{E}_{Z_Y \sim q_{\phi_3}(Z_Y | X, A)}$ as $q_{\phi_1}$, $q_{\phi_2}$, and $q_{\phi_3}$, respectively:

$$\mathbb{E}_{(\epsilon, Z_A, Z_Y) \sim q_\phi(\epsilon, Z_A, Z_Y | X, A, Y)} \left[ \log(p_{\theta_1}(A | X, \epsilon, Z_A)) \right]$$

$$= \mathbb{E}_{q_{\phi_1}} \mathbb{E}_{q_{\phi_2}} \mathbb{E}_{q_{\phi_3}} \left[ \log(p_{\theta_1}(A | X, \epsilon, Z_A)) \right]$$

$$= \mathbb{E}_{q_{\phi_1}} \mathbb{E}_{q_{\phi_2}} \left[ \log(p_{\theta_1}(A | X, \epsilon, Z_A)) \right], \tag{12}$$

and

$$\mathbb{E}_{(\epsilon, Z_A, Z_Y) \sim q_\phi(\epsilon, Z_A, Z_Y | X, A, Y)} \left[ \log(p_{\theta_2}(X | \epsilon, Z_Y)) \right]$$

$$= \mathbb{E}_{q_{\phi_1}} \mathbb{E}_{q_{\phi_2}} \mathbb{E}_{q_{\phi_3}} \left[ \log(p_{\theta_2}(X | \epsilon, Z_Y)) \right]$$

$$= \mathbb{E}_{q_{\phi_2}} \mathbb{E}_{q_{\phi_3}} \left[ \log(p_{\theta_2}(X | \epsilon, Z_Y)) \right], \tag{13}$$

and

$$\mathbb{E}_{(\epsilon, Z_A, Z_Y) \sim q_\phi(\epsilon, Z_A, Z_Y | X, A, Y)} \left[ \log(p_{\theta_3}(Y | X, A, Z_Y)) \right]$$

$$= \mathbb{E}_{q_{\phi_1}} \mathbb{E}_{q_{\phi_2}} \mathbb{E}_{q_{\phi_3}} \left[ \log(p_{\theta_3}(Y | X, A, Z_Y)) \right]$$

$$= \mathbb{E}_{q_{\phi_3}} \left[ \log(p_{\theta_3}(Y | X, A, Z_Y)) \right]. \tag{14}$$

In a similar way, the last term can be also simplified:

$$\mathbb{E}_{(\epsilon, Z_A, Z_Y) \sim q_\phi(\epsilon, Z_A, Z_Y | X, Y, A)} \left[ \log \frac{p(\epsilon)p(Z_A)p(Z_Y)}{q_\phi(\epsilon, Z_A, Z_Y | X, A, Y)} \right]$$

$$= \mathbb{E}_{q_{\phi_1}} \mathbb{E}_{q_{\phi_2}} \mathbb{E}_{q_{\phi_3}} \left[ \log \frac{p(Z_A)p(\epsilon)p(Z_Y)}{q_{\phi_1}(Z_A | X, A)q_{\phi_2}(\epsilon | X, A, Z_Y)q_{\phi_3}(Z_Y | X, A)} \right]$$

$$= \mathbb{E}_{q_{\phi_1}} \left[ \log \frac{p(Z_A)}{q_{\phi_1}(Z_A | X, A)} \right] + \mathbb{E}_{q_{\phi_2}} \mathbb{E}_{q_{\phi_3}} \left[ \log \frac{p(\epsilon)}{q_{\phi_2}(\epsilon | X, A, Z_Y)} \right]$$

$$+ \mathbb{E}_{q_{\phi_3}} \left[ \log \frac{p(Z_Y)}{q_{\phi_3}(Z_Y | X, A)} \right]$$

$$= -kl(q_{\phi_1}(Z_A | X, A) || p(Z_A)) - \mathbb{E}_{q_{\phi_3}} \left[ kl(q_{\phi_2}(\epsilon | X, A, Z_Y) || p(\epsilon)) \right]$$

$$- kl(q_{\phi_3}(Z_Y | X, A) || p(Z_Y)). \tag{15}$$

We combine Eqn. 12, 13, 14, and 15 to get the negative ELBO, i.e., $\mathcal{L}_{\text{ELBO}}$:

$$\mathcal{L}_{\text{ELBO}} = -\mathbb{E}_{Z_A \sim q_{\phi_1}(Z_A | X, A)} \mathbb{E}_{\epsilon \sim q_{\phi_2}(\epsilon | X, A, Z_Y)} \left[ \log(p_{\theta_1}(A | X, \epsilon, Z_A)) \right]$$

$$- \mathbb{E}_{\epsilon \sim q_{\phi_2}(\epsilon | X, A, Z_Y)} \mathbb{E}_{Z_Y \sim q_{\phi_3}(Z_Y | X, A)} \left[ \log(p_{\theta_2}(X | \epsilon, Z_Y)) \right]$$

$$- \mathbb{E}_{Z_Y \sim q_{\phi_3}(Z_Y | X, A)} \left[ \log(p_{\theta_3}(Y | X, A, Z_Y)) \right]$$

$$+ kl(q_{\phi_3}(Z_Y | X, A) || p(Z_Y)) + kl(q_{\phi_1}(Z_A | X, A) || p(Z_A))$$

$$+ \mathbb{E}_{Z_Y \sim q_{\phi_3}(Z_Y | X, A)} \left[ kl(q_{\phi_2}(\epsilon | X, A, Z_Y) || p(\epsilon)) \right]. \tag{16}$$

## B   Details of Model Instantiations

### B.1   Details of regularizing the inference of $Z_A$

We regularize the learned latent graph $\hat{\mathbf{A}}$ based on the prior knowledge that pairs of nodes with high $\gamma$-hop subgraph similarity are more likely to form assortative edges [21, 26, 7], thereby encouraging $\hat{\mathbf{A}}$ to predominantly include such edges.

However, computing $\hat{p}_{ij}$ in every epoch is impractical for large graphs, i.e., $O(N^2)$. To mitigate the issue, we pre-define a candidate graph that consists of the observed edge set $\mathcal{E}$ and a $k$-NN graph based on the $\gamma$-hop subgraph similarity. We denote the set of edges in the $k$-NN graphs as $\mathcal{E}_k^\gamma$. Then, we compute the $\hat{p}_{ij}$ values of the edges in a candidate graph, i.e., $\mathcal{E}_k^\gamma \cup \mathcal{E}$, instead of all edges in $\{(i, j) | i \in \mathcal{V}, j \in \mathcal{V}\}$, to estimate the latent graph structure denoted as $\hat{\mathbf{A}}$. It is important to highlight that obtaining $\mathcal{E}_k^\gamma$ is carried out offline before model training, thus incurring no additional computational overhead during training. This implementation technique achieves a similar effect as minimizing $kl(q_{\phi_1}(Z_A | X, A) || p(Z_A))$ while significantly addressing computational complexity from $O(N^2)$ to $O(|\mathcal{E}_k^\gamma \cup \mathcal{E}|)$, where $N^2 \gg |\mathcal{E}_k^\gamma \cup \mathcal{E}|$.

## C   Further Discussion on DANG

### C.1   Statistical Analysis on Evidence of DANG

To provide empirical evidence of the DANG assumption in addition to intuition, we conduct a statistical analysis on a real-world news network, PolitiFact [34], where node features represent news content, node labels correspond to news topics or categories, and edges denote co-tweet relationships—that is, instances where the same user tweeted both pieces of news. The network includes both fake and benign news, with fake news regarded as feature noise induced by malicious user intent ($\epsilon_X$).

To investigate noise dependency patterns associated with the presence of fake news, We hypothesize that the presence of fake news (i.e., feature noise) leads to noisy graph structures and noisy node labels in news networks. Specifically, we assign a semantic topic to each news article as a node label using k-means clustering over BERT embeddings of the article content. For each node in the graph,

Table 4: Statistics comparison between Fake news and Benign news.

| Category | Mean | 25% | 50% (Median) | 75% |
|---|---|---|---|---|
| Fake news | 1.330 | 1.402 | 1.465 | 1.494 |
| Benign news | 1.084 | 1.004 | 1.353 | 1.431 |
| Mann–Whitney U test (p-value) | | | 4.19e−25 | |

we compute the Shannon entropy of the semantic topic distribution among its neighboring nodes. We then compare these entropy values between fake and benign news nodes.

In Table 4, descriptive statistics reveal that fake news nodes generally exhibit higher entropy than benign news nodes, suggesting that benign news tends to connect to semantically similar articles (homophilic), whereas fake news is more frequently connected to semantically dissimilar articles (heterophilic). This observation aligns with common user behavior: people typically share news related to their interests, whereas fake news is often propagated indiscriminately, regardless of topical relevance [4]. Furthermore, a non-parametric statistical test (Mann–Whitney U test) confirms that the difference in entropy values between fake and benign news is statistically significant. These findings suggest that the presence of fake news (i.e., node feature noise) introduces noisy and heterophilic edges into the graph structure. Furthermore, model-based automated news topic prediction often performs poorly due to noise in both features and graph structures, ultimately resulting in incorrect label annotations.

In summary, these findings empirically support the noisy dependency scenario in real-world scenario where feature noise (i.e., fake news content) can propagate through the graph, generating noisy edges and noisy labels. This highlights the need for our work that explicitly model and mitigate such noise dependencies in real-world networks.

## C.2 Intuitive Examples of DANG

- **User graphs in social networks**: These graphs feature nodes that may represent user's profile or posts, with follow relationship among users defining the graph's structure. The node labels could denote the communities (or interests) of the users. In such scenarios, if users might create fake or incomplete profiles for various reasons, including privacy concerns, some irrelevant users may follow the user based on his/her fake profile, which leads to noisy edges. Moreover, if a user has noisy node features or noisy edges, the user may be assigned to a wrong community (or interest), which leads to noisy labels.

- **User graphs in e-commerce**: Users might create fake or incomplete profiles for various reasons, leading to noisy node features. As a result, products that do not align with the user's genuine interests could be displayed on a web or app page, encouraging the user to view, click on, or purchase these products. Consequently, users are more likely to engage with irrelevant products, leading to a noisy graph structure due to the user's inaccurate features. Moreover, this distortion in users' information and interactions can also alter their associated communities, resulting in noisy node labels.

- **Item graphs in e-commerce**: Fake reviews on products written by a fraudster (i.e., noisy node features) would make other users purchase irrelevant products, which adds irrelevant edges between products (i.e., graph structure noise). Consequently, this would make the automated product category labeling system to inaccurately annotate product categories (i.e., label noise), as it relies on the node features and the graph structure, both of which are contaminated.

- **Item graphs in web graphs**: The content-based features of web pages are corrupted due to poor text extraction or irrelevant information, which leads to noisy node features. In such case, the algorithm responsible for identifying hyperlinks or user navigation patterns might create incorrect or spurious connections between nodes, leading to noisy graph structure. Furthermore, if the features of the nodes are noisy, the algorithms that rely on these features to assign labels (e.g., classifying a web page as a news site or a forum) may result in noisy node labels. Moreover, noises in the graph structure (e.g., incorrect links between web pages) can distort the relational information used by graph-based algorithms, leading to noises in the node labels.

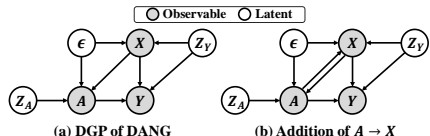

Figure 7: A directed graphical model indicating a DGP of (a) DANG, and (b) the case when adding causal relationship $A \to X$.

- **Item graphs in citation graphs**: In an academic citation network, nodes represent academic papers, edges represent citation relationships, and node features include attributes like title, abstract, authors, keywords, and venue. Recently, generative AI agents have created numerous fake papers with plausible but incorrect attributes, leading to noisy node features. These fake papers get indexed and resemble genuine ones, causing algorithms or researchers to mistakenly create citation links between real and fake papers based on content similarity or keywords, resulting in noisy graph structure. For instance, a well-crafted fake abstract may cause genuine papers to erroneously cite it. Fake papers can corrupt classification algorithms, skewing topic distributions and distorting the citation graph. This affects metrics like citation counts, h-index calculations, and paper influence scores, propagating errors through algorithms that rely on the graph structure, ultimately leading to noisy node labels.

- **Biological Networks**: In addition to the user-item graphs, DANG manifests in the domain of single-cell RNA-sequencing (scRNA-seq). Specifically, in this graph the primary resource is a cell-gene count matrix. A cell-cell graph is commonly employed for downstream tasks, where each cell and its corresponding gene expression are represented as a node and node feature, respectively, and the cell type is considered a node label. However, the node feature, representing gene expression derived from the cell-gene count matrix, often contains noise due to various reasons, such as the dropout phenomenon [35] and batch effect [36]. Since the cell-gene count matrix is the main resource for generating the cell-cell graph [37, 38, 39], such noise acts as a significant obstacle in designing an effective graph structure. Additionally, cell types are annotated using transcripted marker genes, which serve as distinctive features characterizing specific cell types. Noisy node features, therefore, can lead to the misprediction of cell types (node labels). This issue of noise in node features in the biological domain underscores the critical challenge in real-world scenarios.

### C.3 Extension of DANG

While the proposed DANG and DA-GNN demonstrate broader applicability compared to existing methods, they do not perfectly cover all possible noise scenarios. One potential direction to enhance their practicality is to incorporate the causal relationship $X \leftarrow A$, which suggests that graph structure noise can inevitably lead to node feature noise—an occurrence that may manifest in certain real-world scenarios. For instance, consider a social network where node features represent the content to which a user is exposed or interacts with (e.g., views, clicks, or likes), while the graph structure denotes the follower relationships. In such a scenario, if a user follows or is followed by fake accounts, the graph structure might incorporate noisy links (i.e., noisy graph structure). This, in turn, can impact the content to which users are exposed and their interactions (i.e., noisy node features), eventually influencing their community assignments (i.e., noisy node labels). In other words, the noisy node feature and noisy graph structure mutually influence the noise of each other, ultimately incurring the noisy node label. We illustrate its DGP in Fig 7(b). Given that its DGP covers a broader range of noise scenarios that occur in real-world applications than DANG, we expect that directly modeling its DGP has the potential to enhance practical applicability. However, this is a topic we leave for future work.

## D  Further Discussion on DA-GNN

While the implementation of DA-GNN draws inspiration from the spirit of VAE [28] and CausalNL [18], we address complex and unique challenges absent in [28, 18]. Specifically, the incorporation of $A$ necessitates handling supplementary latent variables and causal relationships, such as $Z_A$, $\epsilon_A$, $A \leftarrow \epsilon_A$, $A \leftarrow X$, $Y \leftarrow A$, $A \leftarrow Z_A$, each posing non-trivial obstacles beyond their straightforward extension.

- While [18] assumes that $\epsilon$ only causes $X$, DANG posits that $\epsilon$ also causes $A$, denoted as $A \leftarrow \epsilon_A$. Consequently, DANG requires a novel inference/regularization approach for $\epsilon_A$, which is not addressed in [18], presenting a distinctive technical challenge.

- A simplistic uniform prior is employed to regularize the modeling of $Z_Y$ in [18]. However, upon close examination of the relationship $Y \leftarrow A$, we advocate for a novel regularization approach for $Z_Y$ based on the principle of homophily. This method cannot be elicited through a straightforward application of [18] to the graph.

- By incorporating $A$, $Z_A$, and their associated casualties, we address distinct technical challenges, specifically the inference/regularization of $Z_A$ and the generation of $A$, which cannot be accommodated by a mere extension of [18] to the graph. In particular, we utilize graph structure learning to model $Z_A$, and frame the generation of $A$ as an edge prediction task, incorporating novel regularization techniques for both edge prediction and label. Moreover, we regularize $Z_A$ leveraging our novel prior knowledge to enhance the accuracy and scalability of inference.

We argue that these components are non-trivial to handle through a straightforward application of [18] to the graph domain.

# E    Details on Experimental Settings

## E.1    Datasets

We evaluate DA-GNN and baselines on **five existing datasets** (i.e., Cora [40], Citeseer [40], Amazon Photo and Computers [41]), and ogbn-arxiv [42] and **two newly introduced datasets** (i.e., Amazon Auto and Amazon Garden) that are proposed in this work based on Amazon review data [30, 31] to mimic DANG caused by malicious fraudsters on e-commerce systems (Refer to Appendix E.2.2 for details). The statistics of the datasets are given in Table 5. These seven datasets can be found in these URLs:

- **Cora**: https://github.com/ChandlerBang/Pro-GNN/
- **Citeseer**: https://github.com/ChandlerBang/Pro-GNN/
- **Photo**: https://pytorch-geometric.readthedocs.io/en/latest/
- **Computers**: https://pytorch-geometric.readthedocs.io/en/latest/
- **Arxiv**: https://ogb.stanford.edu/docs/nodeprop/#ogbn-arxiv
- **Auto**: http://jmcauley.ucsd.edu/data/amazon/links.html
- **Garden**: http://jmcauley.ucsd.edu/data/amazon/links.html

Table 5: Statistics for datasets.

| Dataset | # Nodes | # Edges | # Features | # Classes |
|---------|---------|---------|-----------|-----------|
| Cora | 2,485 | 5,069 | 1,433 | 7 |
| Citeseer | 2,110 | 3,668 | 3,703 | 6 |
| Photo | 7,487 | 119,043 | 745 | 8 |
| Computers | 13,381 | 245,778 | 767 | 10 |
| Arxiv | 169,343 | 1,166,243 | 128 | 40 |
| Auto | 8,175 | 13,371 | 300 | 5 |
| Garden | 7,902 | 19,383 | 300 | 5 |

## E.2    Details of Generating DANG

### E.2.1    Synthetic DANG

For the synthetic DANG settings, we artificially generate the noise following the data generation process of the proposed DANG scenario. First, we randomly sample a subset of nodes $\mathcal{V}^{\text{noisy}}$ (i.e., 10%, 30%, and 50% of the whole node set $\mathcal{V}$). To inject node feature noise into the sampled nodes, we randomly flip 0/1 value on each dimension of node features $\mathbf{X}_i$ from Bernoulli distribution with

probability $p = \frac{1}{F}\sum_{i=1}^{F}\mathbf{X}_i$, which results in the noisy features $\mathbf{X}_i^{\text{noisy}}$. After injecting the feature noise, we generate a feature-dependent structure noise (i.e., $A \leftarrow X$) and feature-dependent label noise (i.e., $Y \leftarrow (X, A)$). For the feature-dependent structure noise, we first calculate the similarity vector for each node $v_i$ as $\{s(\mathbf{X}_i^{\text{noisy}}, \mathbf{X}_j)|v_i \in \mathcal{V}^{\text{noisy}}, v_j \in \mathcal{V}\}$ where $s(\cdot, \cdot)$ is a cosine similarity function, and select the node pairs whose feature similarity is top-$k$ highest values. We add the selected node pairs to the original edge set $\mathcal{E}$, which results in $\mathcal{E}^{\text{noisy}}$. To address feature-dependent label noise, we replace the labels of labeled nodes (i.e., training and validation nodes) with randomly sampled labels from a Multinomial distribution, with parameters determined by the normalized neighborhood class distribution. Finally, for the independent structure noise (i.e., $A \leftarrow \epsilon$), we add the randomly selected non-connected node pairs to the $\mathcal{E}^{\text{noisy}}$. Detailed algorithm is provided in Algorithm 2.

### E.2.2  Real-world DANG

We have introduced and released two new graph benchmark datasets, i.e., Auto and Garden, that simulate real-world DANG scenarios on e-commerce systems. To construct these graphs, we utilized metadata and product review data from two categories, "Automotives" and "Patio, Lawn and Garden," obtained from Amazon product review data sources [30, 31]. Specifically, we generated a clean product-product graph where node features are represented using a bag-of-words technique applied to product reviews. The edges indicate co-purchase relationships between products that have been purchased by the same user, and the node labels correspond to product categories. We perform both node classification and link prediction tasks, which are equivalent to categorizing products and predicting co-purchase relationships, respectively.

We simulate the behaviors of fraudsters on a real-world e-commerce platform that incurs DANG. When the fraudsters engage with randomly selected products (i.e., when they write fake product reviews), it would make other users purchase irrelevant products, which introduces a substantial number of malicious co-purchase edges within the graph structure. Additionally, this activity involves the injection of noisy random reviews into the node features. To provide a more detailed description, we designated 100 uers as fraudsters. Furthermore, each of these users was responsible for generating 10 fraudulent reviews in both the Auto and Garden datasets. To generate fake review content, we randomly choose text from existing reviews and duplicate it for the targeted products. This approach guarantees that the fake reviews closely mimic the writing style and content of genuine reviews, while also incorporating irrelevant information that makes it more difficult to predict the product category.

In e-commerce systems, to annotate the node labels (i.e., product categories), machine learning-based automated labeling systems are commonly utilized. Specifically, human annotators manually label a small set of examples, which is used as the training examples to the machine learning model. Subsequently, a machine learning model is trained on these manually labeled product samples to automatically assign categories to other products. Therefore, the systems rely on the information about the products, e.g., reviews of products and co-purchase relationships, to assign categories to products. However, due to the influence of the fraudsters, the noisy node features (i.e., fake product reviews) and noisy graph structure (i.e., co-purchase relationships between irrelevant products) may hinder the accurate assignment of the automated labeling systems, which leads to the noisy node label. To replicate this procedure, we selected 5 examples per category class, which is equivalent to manual labeling process. We then trained a GCN model, leveraging the node features, graph structure, and manually labeled nodes, to predict the true product categories. Consequently, our set of labeled nodes are composed of both manually labeled nodes and nodes labeled using the GCN model. Importantly, the labels of unlabeled nodes were left unchanged and still represented their actual categories. The data generation code is also available at `https://github.com/yeonjun-in/torch-DA-GNN`.

We again emphasize that while existing works primarily focus on the unrealistic noise scenario where graphs contain only a single type of noise, to the best of our knowledge, this is the first attempt to understand the noise scenario in the real-world applications. Furthermore, we propose new graph benchmark datasets that closely imitate a real-world e-commerce system containing malicious fraudsters, which incurs DANG. We expect these datasets to foster practical research in noise-robust graph learning.

### E.3 Baselines

We compare DA-GNN with a wide range of noise-robust GNN methods, which includes feature noise-robust GNNs (i.e., AirGNN [5]), structure-noise robust GNNs (i.e., ProGNN [43], RSGNN [7], STABLE [8] and EvenNet [44]), label noise-robust GNNs (i.e., NRGNN [9] and RTGNN [10]), and multifaceted noise-robust GNNs (i.e., SG-GSR [17]). We also consider WSGNN [20] and GraphGlow [21] that are generative approaches utilizing variational inference technique.

The publicly available implementations of baselines can be found at the following URLs:

- **WSGNN** [20] : https://github.com/Thinklab-SJTU/WSGNN
- **GraphGLOW** [21] : https://github.com/WtaoZhao/GraphGLOW
- **AirGNN** [5] : https://github.com/lxiaorui/AirGNN
- **ProGNN** [43] : https://github.com/ChandlerBang/Pro-GNN
- **RSGNN** [7] : https://github.com/EnyanDai/RSGNN
- **STABLE** [8] : https://github.com/likuanppd/STABLE
- **EvenNet** [44] : https://github.com/Leirunlin/EvenNet
- **NRGNN** [7] : https://github.com/EnyanDai/NRGNN
- **RTGNN** [7] : https://github.com/GhostQ99/RobustTrainingGNN
- **SG-GSR** [17] : https://github.com/yeonjun-in/torch-SG-GSR

### E.4 Evaluation Protocol

We mainly compare the robustness of DA-GNN and the baselines under both the synthetic and real-world feature-dependent graph-noise (DANG). More details of generating DANG is provided in Sec E.2. Additionally, we consider independent feature/structure/label noise, which are commonly considered in prior works in this research field [5, 7, 8, 45, 10]. Specifically, for the feature noise [5], we sample a subset of nodes (i.e., 10%, 30%, and 50%) and randomly flip 0/1 value on each dimension of node features $\mathbf{X}_i$ from Bernoulli distribution with probability $p = \frac{1}{F} \sum_{i=1}^{F} \mathbf{X}_i$. For the structure noise, we adopt the random perturbation method that randomly injects non-connected node pairs into the graph [8]. For the label noise, we generate uniform label noise following the existing works [10, 9].

We conduct both the node classification and link prediction tasks. For node classification, we perform a random split of the nodes, dividing them into a 1:1:8 ratio for training, validation, and testing nodes. Once a model is trained on the training nodes, we use the model to predict the labels of the test nodes. Regarding link prediction, we partition the provided edges into a 7:3 ratio for training and testing edges. Additionally, we generate random negatives that are selected randomly from pairs that are not directly linked in the original graphs. After mode learning with the training edges, we predict the likelihood of the existence of each edge. This prediction is based on a dot-product or cosine similarity calculation between node pairs of test edges and their corresponding negative edges. To evaluate performance, we use Accuracy as the metric for node classification and Area Under the Curve (AUC) for link prediction.

### E.5 Implementation Details

For each experiment, we report the average performance of 3 runs with standard deviations. For all baselines, we use the publicly available implementations and follow the implementation details presented in their original papers.

For DA-GNN, the learning rate is tuned from {0.01, 0.005, 0.001, 0.0005}, and dropout rate and weight decay are fixed to 0.6 and 0.0005, respectively. In the inference of $Z_A$, we use a 2-layer GCN model with 64 hidden dimension as $\text{GCN}_{\phi_1}$ and the dimension of node embedding $d_1$ is fixed to 64. The $\gamma$ value in calculating $\gamma$-hop subgraph similarity is tuned from {0, 1} and $k$ in generating $k$-NN graph is tuned from {0, 10, 50, 100, 300}. In the inference of $Z_Y$, we use a 2-layer GCN model with 128 hidden dimension as $\text{GCN}_{\phi_3}$. In the inference of $\epsilon_X$, the hidden dimension size of $\epsilon_X$, i.e., $d_2$, is fixed to 16. In the inference of $\epsilon_A$, the early-learning phase is fixed to 30 epochs. In

Table 6: Hyperparameter settings on DA-GNN for Table 1.

| Dataset | Setting | lr | $\lambda_1$ | $\lambda_2$ | $\theta_1$ | $k$ | $\gamma$ |
|---|---|---|---|---|---|---|---|
| Cora | Clean | 0.01 | 0.003 | 0.003 | 0.1 | 300 | 1 |
| | DANG-10% | 0.005 | 0.003 | 0.003 | 0.2 | 50 | 1 |
| | DANG-30% | 0.001 | 0.003 | 0.003 | 0.2 | 100 | 1 |
| | DANG-50% | 0.0005 | 30 | 0.003 | 0.3 | 50 | 1 |
| Citeseer | Clean | 0.0005 | 0.003 | 0.3 | 0.1 | 50 | 0 |
| | DANG-10% | 0.005 | 0.3 | 0.003 | 0.3 | 10 | 0 |
| | DANG-30% | 0.001 | 0.003 | 0.003 | 0.1 | 300 | 1 |
| | DANG-50% | 0.001 | 0.003 | 0.003 | 0.1 | 300 | 1 |
| Photo | Clean | 0.01 | 0.03 | 0.3 | 0.1 | 10 | 0 |
| | DANG-10% | 0.0005 | 0.03 | 0.3 | 0.1 | 10 | 0 |
| | DANG-30% | 0.001 | 3 | 0.003 | 0.1 | 10 | 0 |
| | DANG-50% | 0.0005 | 30 | 0.03 | 0.1 | 10 | 0 |
| Comp | Clean | 0.01 | 30 | 0.03 | 0.1 | 10 | 0 |
| | DANG-10% | 0.01 | 0.3 | 0.03 | 0.1 | 10 | 0 |
| | DANG-30% | 0.01 | 0.003 | 0.003 | 0.1 | 10 | 0 |
| | DANG-50% | 0.0005 | 0.003 | 0.03 | 0.1 | 10 | 0 |
| Arxiv | Clean | 0.01 | 0.03 | 0.003 | 0.1 | 0 | 1 |
| | DANG-10% | 0.01 | 0.003 | 0.03 | 0.1 | 0 | 1 |
| | DANG-30% | 0.01 | 0.003 | 0.003 | 0.1 | 0 | 1 |
| | DANG-50% | 0.005 | 3 | 0.03 | 0.1 | 0 | 1 |

the implementation of the loss term $-\mathbb{E}_{Z_A \sim q_{\phi_1}} \mathbb{E}_{\epsilon \sim q_{\phi_2}} [\log(p_{\theta_1}(A|X, \epsilon, Z_A))]$, we tune the $\theta_1$ value from $\{0.1, 0.2, 0.3\}$. In the overall learning objective, i.e., Eqn 4, $\lambda_1$ is tuned from { 0.003, 0.03. 0.3, 3, 30 }, $\lambda_2$ is tuned from { 0.003, 0.03. 0.3 }, and $\lambda_3$ is fixed to 0.001. We report the details of hyperparameter settings in Table 6.

For all baselines, we follow the training instruction reported in their paper and official code. For AirGNN, we tune $\lambda \in \{0.0, 0.2, 0.4, 0.6, 0.8\}$ and set the others as mentioned in the paper for all datasets. For ProGNN, we use the training script reported in the offical code since there are no training guidance in the paper. For RSGNN, we tune $\alpha \in \{0.003, 0.03, 0.3, 3, 30\}$, $\beta \in \{0.01, 0.03, 0.1, 0.3, 1\}$, $n_p \in \{0, 10, 100, 300, 400\}$, and learning rate $\in \{0.01, 0.005, 0.001, 0.0005\}$ for all datasets. For STABLE, We tune $t_1 \in \{0.0, 0.01, 0.02, 0.03, 0.04\}$, $t_2 \in \{0.1, 0.2, 0.3\}$, $k \in \{1, 3, 5, 7, 11, 13\}$, and $\alpha \in \{-0.5, -0.3, -0.1, 0.1, 0.3, 0.6\}$ for all datasets. For EvenNet, we tune $\lambda \in \{0.1, 0.2, 0.5, 0.9\}$ for all datasets following the training script of the official code. For NRGNN, we tune $\alpha \in \{0.001, 0.01, 0.1, 1, 10\}$, $\beta \in \{0.001, 0.01, 0.1, 1, 10, 100\}$, and learning rate $\in \{0.01, 0.005, 0.001, 0.0005\}$ for all datasets. For RTGNN, we tune $K \in \{1, 10, 25, 50, 100\}$, $th_{pse} \in \{0.7, 0.8, 0.9, 0.95\}$, $\alpha \in \{0.03, 0.1, 0.3, 1\}$, and $\gamma \in \{0.01, 0.1\}$, and learning rate $\in \{0.01, 0.005, 0.001, 0.0005\}$. For WSGNN, we use the best hyperparameter setting reported in the paper since there are no training guidance in the paper. For GraphGLOW, we tune learning rate $\in \{0.001, 0.005, 0.01, 0.05\}$, embedding size $d \in \{16, 32, 64, 96\}$, pivot number $P \in \{800, 1000, 1200, 1400\}$, $\lambda \in \{0.1, 0.9\}$, $H \in \{4, 6\}$, $E \in \{1, 2, 3\}$, $\alpha \in \{0, 0.1, 0.15, 0.2, 0.25, 0.3\}$, and $\rho \in \{0, 0.1, 0.15, 0.2, 0.25, 0.3\}$. For SG-GSR, we tune learning rate $\in \{0.001, 0.005, 0.01, 0.05\}$, $\lambda_{\mathcal{E}} \in \{0.2, 0.5, 1, 2, 3, 4, 5\}$, $\lambda_{sp}$ and $\lambda_{fs} \in \{1.0, 0.9, 0.7, 0.5, 0.3\}$, and $\lambda_{aug} \in \{0.1, 0.3, 0.5, 0.7, 0.9\}$.

## F Additional Experimental Results

### F.1 Complexity Analysis

**Theoretical Complexity.** We present a theoretical complexity analysis on training DA-GNN. The computational cost of encoding $Z_Y$ and $\epsilon_X$ is identical to that of GCN and MLP forward pass. The regularization of $Z_Y$ requires $O(c \cdot |\mathcal{E}_k^\gamma \cup \mathcal{E}|)$. Encoding $Z_A$ requires $O(d_1 \cdot |\mathcal{E}_k^\gamma \cup \mathcal{E}|)$, which is significantly reduced by our regularization from $O(d_1 \cdot N^2)$. The computation of encoding $\epsilon_A$ requires $O(d_1 \cdot \mathcal{E})$. Please note that this computation can be ignored since it occurs only during the early learning phase. Decoding $A$ requires $O(|\mathcal{E} + \mathcal{E}^-|)$. Decoding $X$ and $Y$ requires MLP and GCN forward pass. The primary computational burden stems from the encoding $Z_A$ and decoding $A$. Our regularization technique has alleviated this computational load, making DA-GNN more scalable.

**Large Scale Graph.** To demonstrate the scalability of DA-GNN, we consider a larger graph dataset, ogbn-arxiv [42]. Table 1 clearly illustrates that DA-GNN exhibits superior scalability and robustness in comparison to other baseline methods.

Table 7: Training time comparison on Cora dataset under DANG 50%.

| Training time | AirGNN | ProGNN | RSGNN | STABLE | EvenNet | NRGNN | RTGNN | SG-GSR | DA-GNN |
|---|---|---|---|---|---|---|---|---|---|
| Total (sec) | 20.9 | 702.1 | 159.9 | 53.3 | **0.8** | 100.3 | 118.7 | 86.3 | 46.3 |
| per epoch (sec) | 0.04 | 1.77 | 0.16 | - | **0.004** | 0.20 | 0.18 | 0.11 | 0.09 |

**Training Time Comparison** We compare the training time of DA-GNN with the existing noise robust graph learning baselines to analyze the computational complexity of DA-GNN . In Table 7, we report the total training time and training time per epoch on Cora with DANG 50%. Note that since STABLE is a 2-stage method, we did not report the training time per epoch. The results show that DA-GNN requires significantly less total training time and training time per epoch compared to ProGNN, RSGNN, STABLE, NRGNN, RTGNN, and SG-GSR. This suggests that DA-GNN's training procedure is faster than that of most baselines while still achieving substantial performance improvements. Although AirGNN and EvenNet require much less training time than DA-GNN, their node classification accuracy is notably worse than other methods, including DA-GNN. This indicates that, despite their fast training times, they may not be suitable for real-world deployments. In summary, DA-GNN demonstrates superior performance compared to the baselines while maintaining acceptable training times.

### F.2 Sensitivity Analysis

We analyze the sensitivity of the coefficient $\lambda_1$ and $\lambda_2$ in Eqn 4, $\theta$, and $\gamma$. To be specific, we increase $\lambda_1$ value from $\{0.0, 0.003, 0.03, 0.3, 3\}$, $\lambda_2$ value from $\{0.0, 0.003, 0.03, 0.3\}$, $\theta$ from $\{0.1, 0.2, 0.3\}$, and $\gamma$ from $\{0, 1\}$. We then evaluate the node classification accuracy of DA-GNN under DANG.

- In Fig 8(a) and 9(a), we notice that DA-GNN consistently surpasses the state-of-the-art baseline, EvenNet, regardless of the $\lambda_1$ value, demonstrating the robustness of DA-GNN. Furthermore, we observe that the performance significantly drops when $\lambda_1 = 0$. This highlights the importance of modeling the causal relationship $A \leftarrow (X, \epsilon, Z_A)$ for robustness under DANG, as $\lambda_1$ is directly related to the loss term $\mathcal{L}_{\text{edge-rec}}$, i.e., $-\mathbb{E}_{Z_A}\mathbb{E}_\epsilon\left[\log(p_{\theta_1}(A|X, \epsilon, Z_A))\right]$.

- In Fig 8(b) and 9(b), we observe that DA-GNN generally outperforms the sota baseline regardless of the value of $\lambda_2$, indicating the stability of DA-GNN. Moreover, we can see a performance decrease when $\lambda_2 = 0$. This observation suggests that the regularization on the inferred latent node label $Z_Y$ using the inferred latent structure $Z_A$ effectively handles the noisy labels. This conclusion is drawn from the fact that $\lambda_2$ is directly linked to the loss term $\mathcal{L}_{\text{hom}}$, i.e., $kl(q_{\phi_3}(Z_Y|X, A)||p(Z_Y))$.

- In Fig 8(c) and 9(c), we analyze the hyperparameter sensitivity of $k$ and observe that $k$ plays a critical role and requires some tuning. To recap the role of $k$, we pre-define a proxy graph based on subgraph similarity, where each node connects to $k$ neighbors. We then compute $\hat{p}$ as the edge weights on this proxy graph, which corresponds to the regularization term minimizing $kl(q_{\phi_1}(Z_A|X, A)||p(Z_A)$. Sensitivity to $k$ highlights the importance of accurately inferring the latent graph structure $Z_A$. This is expected, as using rich neighborhood information from $Z_A$ enables robust message passing, which helps mitigate noise in the observed graphs. We restrict the search to just five values: $\{0, 10, 50, 100, 300\}$. This narrow range consistently yielded effective performance across all seven datasets, suggesting that tuning $k$ is not overly burdensome.

- In Fig 8(d) and 9(d), we observe that DA-GNN consistently outperforms the state-of-the-art baseline, EvenNet, across all values of $\theta$, demonstrating the robustness of the prediction regularization method in Eqn3.

- In Fig 8(e) and 9(e), we observe that DA-GNN consistently surpasses the state-of-the-art baseline, EvenNet, across all values of $\gamma$, emphasizing the stability of regularizing the inferred $Z_A$ in modeling $q_{\phi_1}(Z_A|X, A)$.

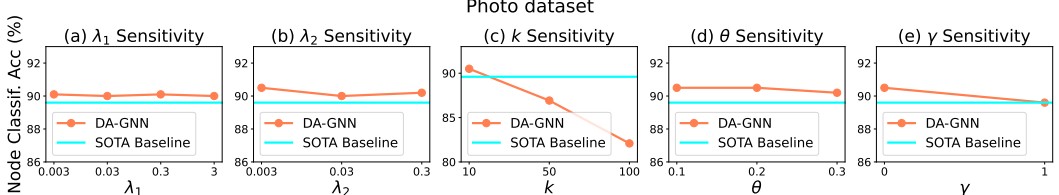

Figure 8: Sensitivity analysis on $\lambda_1$, $\lambda_2$, $\theta$, and $\gamma$. We conduct the experiments on Photo dataset under DANG–30%

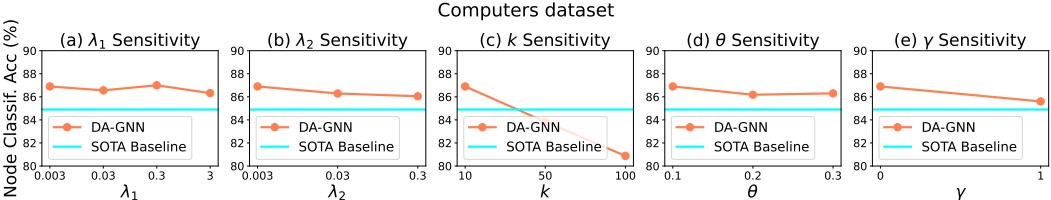

Figure 9: Sensitivity analysis on $\lambda_1$, $\lambda_2$, $\theta$, and $\gamma$. We conduct the experiments on Computers dataset under DANG–30%

## F.3 Robustness Evaluation under Variants of DANG

### F.3.1 Variants of Synthetic DANG

In the generation process of our synthetic DANG, we have three variables: 1) the overall noise rate, 2) the amount of noise dependency ($X \to A$, $X \to Y$, $A \to Y$), and 3) the amount of independent structure noise ($\epsilon \to A$).

Table 8: Node classification results on DANG with increased noise dependency

| Dataset | Setting | AirGNN | RSGNN | STABLE | EvenNet | NRGNN | RTGNN | SG-GSR | DA-GNN |
|---------|---------|--------|-------|--------|---------|-------|-------|--------|--------|
| Cora | DANG-10% | 78.3±0.3 | 79.0±0.1 | 79.5±0.4 | 76.9±1.2 | 78.6±0.5 | 78.8±0.5 | 78.5±0.2 | **79.8±0.2** |
| | DANG-30% | 57.6±0.5 | **67.9±0.6** | 65.2±1.4 | 55.8±1.3 | 63.8±0.9 | 66.1±0.6 | 56.9±0.7 | 67.0±0.3 |
| | DANG-50% | 40.1±0.5 | 49.4±0.9 | 45.7±1.2 | 40.5±1.0 | 47.5±0.5 | 48.1±0.8 | 40.1±1.2 | **51.6±0.9** |
| Citeseer | DANG-10% | 65.7±1.1 | **72.9±0.4** | 68.4±0.7 | 68.8±0.6 | 69.7±1.0 | 69.8±0.0 | 70.3±0.5 | 72.7±0.3 |
| | DANG-30% | 57.2±0.9 | 63.3±0.6 | 57.2±0.1 | 57.2±0.5 | 59.6±0.7 | 60.1±0.7 | 62.0±0.9 | **64.9±0.6** |
| | DANG-50% | 39.8±0.7 | 49.4±1.0 | 41.3±1.8 | 42.2±0.5 | 42.9±0.6 | 43.7±0.7 | 46.1±1.3 | **51.4±0.2** |

- For the first variable, our experiments already addressed it by varying the noise rate from 0 to 50.

- For the second variable, we conduct an additional analysis by substantially increasing or decreasing the degree of noise dependency. Specifically, we increase the number of structure noise edges caused by feature noise by approximately 4×, and similarly amplify the amount of label noise induced by both feature and structure noise by 4×. We also evaluate a setting where noise dependencies are completely removed—this corresponds to a scenario with independent feature and structure noise. As shown in Table 8 and Table 9, DA-GNN consistently outperforms all baselines on the strong presence of noise dependency, and shows competitive performance on the weak presence of noise dependency.

- For the third variable, we perform an additional analysis by doubling the amount of independent structure noise. We also evaluate the case where no independent structure noise is present. As shown in Table 10 and Table 11, DA-GNN consistently outperforms all baselines across both settings.

These results demonstrate that DA-GNN consistently outperforms other baselines under varying degrees of DANG, highlighting its practical applicability across diverse real-world noise conditions.

Table 9: Node classification results on DANG without noise dependency

| Dataset | Setting | AirGNN | RSGNN | STABLE | EvenNet | NRGNN | RTGNN | SG-GSR | DA-GNN |
|---|---|---|---|---|---|---|---|---|---|
| Cora | DANG-10% | 83.1±0.4 | 83.9±0.6 | 83.9±0.4 | 84.0±0.5 | 83.8±0.2 | **84.8±0.2** | **84.8±0.1** | **84.8±0.1** |
| | DANG-30% | 77.7±0.5 | 79.9±0.4 | 76.8±0.6 | 74.8±0.6 | 77.8±0.8 | 80.0±0.1 | **80.1±0.1** | **80.1±0.3** |
| | DANG-50% | 66.9±2.6 | 72.7±0.6 | 70.3±1.8 | 61.3±3.4 | 69.1±0.8 | 73.2±0.6 | 72.0±0.2 | **75.4±0.0** |
| Citeseer | DANG-10% | 68.8±0.3 | **75.9±0.9** | 71.9±0.6 | 74.0±0.3 | 74.1±0.7 | 74.4±0.4 | 75.5±0.5 | **75.9±0.4** |
| | DANG-30% | 63.6±0.2 | 70.7±0.5 | 67.3±0.2 | 67.9±0.3 | 69.7±0.3 | 69.4±0.6 | **72.0±0.4** | 71.5±0.3 |
| | DANG-50% | 59.7±0.7 | 64.6±0.7 | 59.3±0.6 | 61.3±0.6 | 63.4±0.4 | 64.6±0.2 | **66.4±0.3** | 64.7±0.6 |

Table 10: Node classification results on DANG without independent structure noise

| Dataset | Setting | AirGNN | RSGNN | STABLE | EvenNet | NRGNN | RTGNN | SG-GSR | DA-GNN |
|---|---|---|---|---|---|---|---|---|---|
| Cora | DANG-10% | 81.1±0.5 | 81.1±0.6 | 83.1±0.7 | 81.4±0.3 | 82.1±0.3 | 82.6±0.2 | 82.5±0.1 | **83.9±0.3** |
| | DANG-30% | 73.9±1.7 | 73.6±0.3 | 76.9±0.3 | 69.7±0.7 | 76.3±0.4 | 74.8±0.9 | 77.7±0.3 | **79.6±0.6** |
| | DANG-50% | 64.6±2.3 | 60.3±1.3 | 66.4±0.6 | 51.6±0.5 | 64.4±1.0 | 62.8±0.6 | 69.5±1.0 | **72.1±0.4** |
| Citeseer | DANG-10% | 68.3±0.6 | 71.8±0.7 | 72.4±0.9 | 72.8±0.1 | 73.1±0.3 | 73.7±0.2 | 74.5±0.4 | **74.7±0.1** |
| | DANG-30% | 58.5±0.5 | 63.5±0.9 | 64.6±0.2 | 63.1±0.4 | 64.3±1.4 | 64.8±0.9 | 66.1±0.6 | **66.4±0.6** |
| | DANG-50% | 54.3±0.2 | 55.9±0.3 | 58.1±1.0 | 51.2±2.1 | 56.7±0.2 | 56.6±0.9 | 59.3±0.6 | **60.3±1.2** |

#### F.3.2 Variants of Real-world DANG

We conduct an experiment where we independently double each of the following: (1) the number of fraudsters (i.e., nodes with noisy features) and (2) the activeness of fraudsters (i.e., the amount of structure noise they introduce) in our real-world DANG generation process. As a result, label noise also increases accordingly, in proportion to the amount of generated feature and structure noise.

As shown in Table 12, DA-GNN demonstrates competitive performance and, in many cases, outperforms other baselines under these intensified noise conditions.

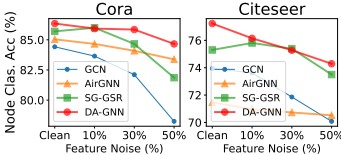

Figure 10: Node classification accuracy under independent feat. noise.

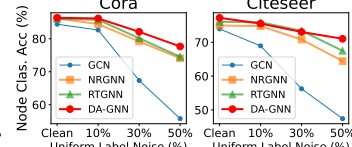

Figure 11: Node classification accuracy under independent stru. noise.

Figure 12: Node classification accuracy under independent label noise.

#### F.4 Qualitative Analysis

In Fig 13(b), we analyze the inference of $Z_A$ by comparing the distribution of $\hat{p}_{ij}$ values, which constitute the estimated latent graph structure $\hat{\mathbf{A}}$, between noisy edges and the original clean edges. It is evident that the estimated edge probabilities $\hat{p}_{ij}$ for noisy edges are predominantly assigned smaller values, while those for clean edges tend to be assigned larger values. It illustrates DA-GNN effectively mitigates the impact of noisy edges during the message-passing process, thereby enhancing its robustness in the presence of noisy graph structure. This achievement can be attributed to the label regularization effect achieved through the accurate inference of $\epsilon_A$. Specifically, as the observed graph structure contains noisy edges, the inaccurate supervision for $\mathcal{L}_{\text{rec-edge}}$ impedes the distinction between noisy edges and the original clean edges in terms of edge probability values $\hat{p}_{ij}$. However, the label regularization technique proves crucial for alleviating this issue, benefitting from the accurate inference of $\epsilon_A$.

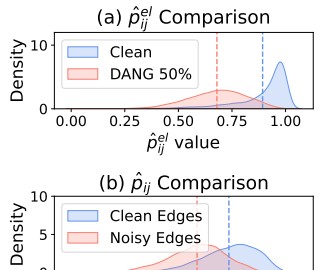

Figure 13: (a) Distribution of $\hat{p}_{ij}^{el}$ values. (b) Distribution of $\hat{p}_{ij}$ values under DANG-50%. Dashed lines are averages. Cora dataset is used.

Table 11: Node classification results on DANG with increased independent structure noise

| Dataset | Setting | AirGNN | RSGNN | STABLE | EvenNet | NRGNN | RTGNN | SG-GSR | DA-GNN |
|---|---|---|---|---|---|---|---|---|---|
| Cora | DANG-10% | 78.9±0.7 | 81.6±0.3 | 80.9±0.5 | 78.9±0.3 | 79.9±0.4 | 81.8±0.3 | 81.4±0.2 | **82.5±0.2** |
| | DANG-30% | 66.1±1.8 | 70.6±0.9 | 72.0±0.8 | 61.0±0.9 | 72.2±0.6 | 70.1±0.6 | 72.4±0.2 | **75.4±0.4** |
| | DANG-50% | 47.3±0.5 | 56.7±0.2 | 57.9±1.6 | 42.1±2.1 | 58.8±0.6 | 55.6±0.5 | 61.2±1.5 | **65.4±0.6** |
| Citeseer | DANG-10% | 66.5±0.4 | **74.0±0.3** | 70.8±0.4 | 70.3±0.8 | 71.9±0.1 | 72.6±0.3 | 72.6±0.4 | 73.6±0.2 |
| | DANG-30% | 58.0±0.2 | 63.8±0.6 | 62.3±1.8 | 60.1±0.3 | 61.6±0.9 | 63.2±0.5 | 63.7±0.8 | **65.1±0.5** |
| | DANG-50% | 49.4±0.6 | 55.1±0.3 | 51.7±1.7 | 45.9±0.7 | 50.8±0.8 | 52.0±1.1 | 54.2±0.2 | **55.4±1.2** |

Table 12: Node classification results on variants of real-world DANG

| Dataset | Setting | AirGNN | RSGNN | STABLE | EvenNet | NRGNN | RTGNN | DA-GNN |
|---|---|---|---|---|---|---|---|---|
| Auto | DANG w/ doubled # frauds | 54.6±1.5 | 53.4±0.7 | 55.4±0.1 | 56.5±0.6 | 55.9±1.5 | 54.3±2.7 | **60.1±0.7** |
| | DANG w/ doubled structure noise | 56.9±0.7 | 50.9±0.6 | **58.1±2.0** | 53.6±2.2 | 55.8±1.3 | 56.78±0.8 | 55.6±1.0 |
| Garden | DANG w/ doubled # fraudsters | 57.1±1.3 | 65.0±0.5 | 69.8±2.3 | 69.3±1.8 | 70.8±0.7 | 70.6±0.9 | **71.9±0.6** |
| | DANG w/ doubled structure noise | 69.9±2.8 | 69.4±1.3 | 72.0±0.5 | 72.4±0.9 | 71.0±2.4 | **75.3±0.4** | 74.4±0.2 |

We qualitatively analyze how well DA-GNN infers the latent variables $\epsilon_A$ and $Z_A$. In Fig 13(a), we investigate the inference of $\epsilon_A$ by comparing the distribution of $\hat{p}_{ij}^{el}$ values estimated during training on clean and noisy graphs (DANG-50%). We observe that $\hat{p}_{ij}^{el}$ values estimated from the clean graph tend to be close to 1, while those from the graph with DANG are considerably smaller. It suggests the inference of $\epsilon_A$ was accurate, as the high values of $\hat{p}_{ij}^{el}$ indicate that the model recognizes the edge $(i, j)$ as a clean edge.

Furthermore, to verify the distinction between the noisy and clean scenarios, We conduct a non-parametric analysis, Mann–Whitney U test, which require no distributional assumptions. The results are as follows:

- Fig 13(a): Statistic=62337852.0, p-value=0.0
- Fig 13(b): Statistic=40277922.0, p-value=0.0.

Note that we found the scipy.stats package displays p-values as zero when they are extremely low. Therefore, we reported the corresponding test statistics with p-values. The results indicate highly significant differences between the groups.

## F.5 Comparison with the Naive Combination of Existing Works

So far, we have observed that existing approaches fail to generalize to DANG since they primarily focus on graphs containing only a single type of noise. A straightforward solution might be to naively combine methods that address each type of noise individually. To explore this idea, we consider AirGNN as the feature noise-robust GNN (*FNR*), RSGNN as the structure noise-robust GNN (*SNR*), and RTNN as the label noise-robust GNN (*LNR*). We carefully implement all possible combinations among *FNR*, *SNR*, and *LNR*.

In Table 13, we observe that naive combination can improve robustness in some cases, but it may not consistently yield favorable results. For example, combining *FNR* and *SNR* notably enhances robustness. However, when we combine all three (*FNR*, *SNR*, and *LNR*), which is expected to yield the best results, performance even decreases. This could be attributed to compatibility issues among the methods arising from the naive combination. Furthermore, although some combinations improve robustness, DA-GNN consistently outperforms all combinations. We attribute this to the fact that naively combining existing methods may not capture the causal relationships in the DGP of DANG, limiting their robustness. In contrast, DA-GNN successfully captures these relationships, resulting in superior performance.

Table 13: Comparison with the naive combination of existing noise-robust graph learning methods. *FNR*, *SNR*, and *LNR* denote the feature noise-robust, structure noise-robust, and label noise-robust graph learning methods, respectively. We consider AirGNN as *FNR*, RSGNN as *SNR*, and RTGNN as *LSR* methods.

| Component | | | Cora | | | | Citeseer | | | |
|---|---|---|---|---|---|---|---|---|---|---|
| *FNR* | *SNR* | *LNR* | Clean | DANG 10% | DANG 30% | DANG 50% | Clean | DANG 10% | DANG 30% | DANG 50% |
| ✓ | ✗ | ✗ | 85.0±0.2 | 79.7±0.5 | 71.5±0.8 | 56.2±0.8 | 71.5±0.2 | 66.2±0.7 | 58.0±0.4 | 50.0±0.6 |
| ✗ | ✓ | ✗ | 86.2±0.5 | 81.9±0.3 | 71.9±0.5 | 58.1±0.2 | 75.8±0.4 | 73.3±0.5 | 63.9±0.5 | 55.3±0.4 |
| ✗ | ✗ | ✓ | 86.1±0.2 | 81.6±0.5 | 72.1±0.6 | 60.8±0.4 | 76.1±0.4 | 73.2±0.2 | 63.5±2.1 | 54.2±1.8 |
| ✓ | ✓ | ✗ | 86.0±0.3 | 82.0±0.3 | 75.0±0.8 | 68.8±0.6 | 75.1±0.8 | 73.1±0.6 | 63.6±0.8 | 57.8±0.8 |
| ✓ | ✗ | ✓ | 85.2±0.7 | 70.1±0.1 | 56.7±0.4 | 48.0±0.5 | 75.8±0.5 | 72.3±0.3 | 59.0±0.7 | 49.0±0.2 |
| ✗ | ✓ | ✓ | 85.0±0.2 | 79.4±0.9 | 72.3±0.5 | 63.0±0.4 | 76.7±0.3 | **74.3±0.9** | 64.8±0.3 | 55.3±0.5 |
| ✓ | ✓ | ✓ | **86.3±0.3** | 82.4±0.3 | 67.0±0.9 | 53.6±0.6 | 76.6±0.2 | 73.0±0.7 | 64.1±0.2 | 52.7±1.1 |
| DA-GNN | | | 86.2±0.7 | **82.9±0.6** | **78.2±0.3** | **69.7±0.6** | **77.3±0.6** | **74.3±0.9** | **65.6±0.6** | **59.0±1.8** |

---

**Algorithm 1** Training Algorithm of DA-GNN.

1: **Input**: Observed graph $\mathcal{G} = \langle \mathcal{V}, \mathcal{E} \rangle$, node feature $\mathbf{X} \in \mathbb{R}^{N \times F}$, node label $\mathbf{Y} \in \mathbb{R}^{N \times C}$
2: Initialize trainable parameters $\phi_1, \phi_2, \phi_3, \theta_2, \theta_3$
3: Initialize $\hat{p}_{ij}^{el}$ to one vector $\mathbf{1}$.
4: Generate a $k$-NN graph $\mathcal{E}_k^\gamma$ based on the $\gamma$-hop subgraph similarity
5: Pre-define a candidate graph by $\mathcal{E}_k^\gamma \cup \mathcal{E}$
6: **while** *not converge* **do**
7:     /* Inference of $Z_A$ */
8:     Feed $\mathbf{X}$ and $\mathbf{A}$ to $\text{GCN}_{\phi_1}$ to obtain the node embeddings $\mathbf{Z}$
9:     Calculate the $\hat{p}_{ij}$ on the candidate graph $\mathcal{E}_k^\gamma \cup \mathcal{E}$ based on $\mathbf{Z}$ to obtain $\hat{\mathbf{A}}$.
10:     /* Inference of $Z_Y$ */
11:     Feed $\mathbf{X}$ and $\hat{\mathbf{A}}$ to $\text{GCN}_{\phi_3}$ to get $\hat{\mathbf{Y}}$
12:     /* Inference of $\epsilon_X$ */
13:     Feed $\mathbf{X}$ and $\hat{\mathbf{Y}}$ to the $\text{MLP}_{\phi_2}$ to get node embeddings that follow $\mathcal{N}(\mathbf{0}, \mathbf{I})$
14:     /* Inference of $\epsilon_A$ */
15:     **if** early-learning phase **then**
16:        $\hat{p}_{ij}^c \leftarrow \rho(s(\mathbf{Z}_i, \mathbf{Z}_j))$
17:        $\hat{p}_{ij}^{el} \leftarrow \xi \hat{p}_{ij}^{el} + (1 - \xi)\hat{p}_{ij}^c$
18:        Convert $\hat{p}_{ij}^{el}$ into $\tau_{ij}$
19:     **end if**
20:     /* Generation of $A$ */
21:     Obtain an edge prediction $w_{ij} = \theta_1 \hat{p}_{ij} + (1 - \theta_1)s(\mathbf{X}_i, \mathbf{X}_j)$
22:     /* Generation of $X$ */
23:     Obtain the reconstruction of node features based on decoder $\text{MLP}_{\theta_2}$ and its input $\epsilon_X$ and $\hat{\mathbf{Y}}$.
24:     /* Generation of $Y$ */
25:     Obtain node prediction $\hat{\mathbf{Y}}_{\text{dec}}$ based on classifier $\text{GCN}_{\theta_3}$ and its input $\mathbf{X}$ and $\mathbf{A}$.
26:     /* Loss calculation */
27:     Calculate the objective function $\mathcal{L}_{\text{cls-enc}} + \lambda_1 \mathcal{L}_{\text{rec-edge}} + \lambda_2 \mathcal{L}_{\text{hom}} + \lambda_3 (\mathcal{L}_{\text{rec-feat}} + \mathcal{L}_{\text{cls-dec}} + \mathcal{L}_{\text{p}})$.
28:     /* Parameter updates */
29:     Update the parameters $\phi_1, \phi_2, \phi_3, \theta_2, \theta_3$ to minimize the overall objective function.
30: **end while**
31: **Return**: learned model parameters $\phi_1, \phi_2, \phi_3, \theta_2, \theta_3$

---
**Algorithm 2** Data Generation Algorithm of Synthetic DANG.
---
1: **Input**: Clean graph $\mathcal{G} = \langle \mathcal{V}, \mathcal{E} \rangle$, node feature $\mathbf{X} \in \mathbb{R}^{N \times F}$, node label $\mathbf{Y} \in \mathbb{R}^{N \times C}$, noise rate $\eta\%$
2: /* Injection of feature noise */
3: $\mathcal{V}^{\text{noisy}} \leftarrow$ Randomly sample a $\eta\%$ subset of nodes
4: $\mathbf{X}^{\text{noisy}} \leftarrow \mathbf{X}$
5: **for** $v_i$ in $\mathcal{V}^{\text{noisy}}$ **do**
6:     $p_i \leftarrow \frac{1}{F} \sum_{j=1}^{F} \mathbf{X}_{ij}$
7:     **for** $j \leftarrow 1$ to $F$ **do**
8:         $\mathbf{X}_{ij}^{\text{noisy}} \leftarrow \textbf{BernoulliSample}(p_i)$
9:     **end for**
10: **end for**
11: /* Injection of feature-dependent structure noise */
12: $\mathcal{E}^{\text{noisy}} \leftarrow \mathcal{E}$
13: **for** $v_i$ in $\mathcal{V}^{\text{noisy}}$ **do**
14:     $\mathbf{s} \leftarrow \mathbf{0} \in \mathbb{R}^N$
15:     **for** $j \leftarrow 1$ to $N$ **do**
16:         $\mathbf{s}_j \leftarrow s(\mathbf{X}_i^{\text{noisy}}, \mathbf{X}_j)$
17:     **end for**
18:     Append $k$ pairs of nodes with the highest $\mathbf{s}$ values to $\mathcal{E}^{\text{noisy}}$
19: **end for**
20: /* Injection of feature-dependent label noise */
21: $\mathbf{Y}^{\text{noisy}} \leftarrow \mathbf{Y}$
22: **for** $v_i$ in $\mathcal{V}^L$ **do**
23:     **if** $v_i$ has noisy feature or noisy structure **then**
24:         $\mathbf{p}_i \leftarrow$ Obtain normalized neighborhood class distribution of node $v_i$
25:         $\mathbf{Y}_i^{\text{noisy}} \leftarrow \textbf{MultinomialSample}(\mathbf{p}_i)$
26:     **end if**
27: **end for**
28: /* Injection of independent structure noise */
29: Randomly append pairs of nodes to $\mathcal{E}^{\text{noisy}}$
30: **Return:** noisy graph $\mathcal{G} = \langle \mathcal{V}, \mathcal{E}^{\text{noisy}} \rangle$, noisy node feature $\mathbf{X}^{\text{noisy}}$, noisy node label $\mathbf{Y}^{\text{noisy}}$
---

