# OpenReview forum: "Training Robust Graph Neural Networks by Modeling Noise Dependencies"
_NeurIPS.cc/2025/Conference — NeurIPS 2025 poster_

### Official Review · Reviewer_KkeS · 2025-06-06

**Clarity:** 3
**Significance:** 2
**Originality:** 2
**Rating:** 4
**Confidence:** 3

**Summary:**

This paper proposes DA-GNN, a robust graph neural network model designed to handle Dependency-Aware Noise on Graphs (DANG)—a realistic scenario where noise in node features affects both graph structure and labels. Unlike prior work assuming independent noise, DA-GNN models the full causal noise dependencies using a deep generative framework with variational inference. The authors introduce new benchmarks to simulate DANG and demonstrate that DA-GNN consistently outperforms existing robust GNNs across various noise types, including synthetic, real-world, and extreme noise scenarios.

**Questions:**

- How representative is the DANG assumption across diverse real-world domains beyond social networks and e-commerce? Can you provide empirical evidence—such as statistical analysis or real-world data studies, rather than intuition—that supports the prevalence of the causal relationships assumed in DANG in domains like biological networks, citation graphs, or recommendation systems? Without such support, the synthetic DANG settings constructed on citation networks in the experimental section seem to be an artificial formulation tailored to fit the proposed model.
- The "Auto" and "Garden" datasets are generated using specific heuristics for fake reviews and label noise. How sensitive are your results to these design choices? Could the results be skewed by the data generation pipeline favoring DA-GNN?
- DA-GNN involves multiple encoders, decoders, and regularization terms. What is the actual cost in terms of runtime/memory, especially for large-scale graphs such as ogbn-arxiv? How sensitive is DA-GNN to hyperparameter settings in low-resource settings?
- In Fig.3, the font is too small and the layout is overly compact. It is recommended to stretch the figure horizontally and appropriately increase the font size. In addition, in the Inference Encoders section, is there a missing arrow from $Z_A$ to Noise Env. Infer.?
- To what extent does the proposed method rely on the specific GNN architectures used in the encoder and decoder modules? For example, the encoders for inferring $Z_A$ and $Z_Y$, use GCNs, and the decoder for label prediction also adopts a GCN. Have the authors evaluated how performance is affected when substituting these GNN components with alternative architectures (e.g., GraphSAGE, GAT)? This would help clarify whether DA-GNN's robustness primarily stems from the causal modeling framework itself or from the specific choice of backbone GNNs. Additionaly, The low-pass filtering nature of GCN inherently imposes a homophily inductive bias on the encoding/decoding process. Could this lead to poor performance of the model on heterophilous graph datasets?
- How does DA-GNN perform under varying levels of noise correlation or dependency strength in the DANG setting? While the paper evaluates robustness under different noise rates (10%, 30%, 50%), it is unclear how sensitive the model is to the strength of the dependency between feature noise and structure/label noise. For example, what happens if the correlation between noisy features and added edges is weak or inconsistent? A controlled study varying the dependency strength (rather than just the noise rate) would better validate the assumptions underlying DANG.

**Ethical Concerns:**

["NO or VERY MINOR ethics concerns only"]

**Final Justification:**

The rebuttal addressed my concerns. Based on this, I have decided to raise my rating.

**Limitations:**

Yes

**Quality:**

2

**Strengths And Weaknesses:**

**Strengths:**
- The paper is technically sound and presents a complete solution pipeline: from motivation and formalization of the DANG (Dependency-Aware Noise on Graphs) scenario to the proposed DA-GNN model and empirical evaluation.
- The use of a variational inference framework to model noise dependencies in node features, structure, and labels is well-motivated and backed by a full derivation.
- Experiments are extensive, covering both synthetic and real-world datasets, and include comparisons across a wide range of baselines and noise types.


**Weaknesses:**
- The paper has several writing issues that hinder clarity. Some symbols in the equations are not clearly defined—for example, the meanings of $q_\phi$, $p_\theta$, $p(\epsilon)$, $p(Z_Y)$ in Eq.1, as well as the use of Bern(.) in line 210, are not explicitly explained in the main text. Some methodological details are buried in the appendix but are critical to understanding.
- Insufficient introduction of background and related work. The paper assumes familiarity with graphical causal modeling and variational inference, which might limit accessibility for a broader NeurIPS audience.
- While the problem formulation is original, the method borrows heavily from existing variational techniques (e.g., VAE, CausalNL), and the architectural components (e.g., GCN-based encoders, MLP decoders) are standard. The generative modeling perspective is not entirely new in GNN literature—what is new is its application to a more realistic noise model, which, while valuable, limits the novelty of the method itself.
- Motivation, methodological details and experimental phenomena require further explanation (see the Questions section).

---

> ### Author Rebuttal · Authors · 2025-07-31
>
> We sincerely thank the reviewer for carefully reading our paper and providing thoughtful and constructive feedback. We believe the comments are highly valuable and have significantly helped us improve the quality and clarity of the manuscript.
>
> ---
>
> ## Response to W1
>
> In the revised manuscript, we have carefully revised the main text to improve readability and clarity. Specifically:
> - We will provide clear definitions for all probabilistic terms in Eq. (1), directly in the main text.
> - We will explicitly explain the use of Bern(⋅) to in main text.
> - Several important methodological details that were previously located in the appendix will be moved into the main text to ensure the paper is more self-contained and easier to follow.
>
> We believe these revisions significantly improve the clarity and accessibility of the paper, and we thank the reviewer for raising this important point.
>
> ---
>
> ## Response to W2
>
> We agree that the original submission assumed a certain level of familiarity with graphical causal modeling and variational inference. In the revised manuscript, we have added the background sections in Appendix to provide a more accessible overview of the necessary concepts.
>
> ---
>
> ## Response to W3
>
> While the implementation of DAGNN draws inspiration from the spirit of VAE  and CausalNL [1], we address complex and unique challenges absent in them. Specifically, the incorporation of $A$ necessitates handling supplementary latent variables and causal relationships, such as $Z_A$, $\epsilon_A$, $A$ ← $\epsilon_A$, $A$ ← $X$, $Y$ ← $A$, $A$ ← $Z_A$, each posing non-trivial obstacles beyond their straightforward extension to the graph domain. It's worth noting that we have addressed this issue in Appendix D to underscore the technical advancements of DA-GNN in comparison to [1].
>
>
> [1] Instance-dependent Label-noise Learning under a Structural Causal Model
>
>
> ---
>
> ## Response to Q1
>
> We conduct a statistical analysis on a real-world news network, PolitiFact [1], where node features represent news content, node labels correspond to news topics or categories, and edges denote co-tweet relationships—that is, instances where the same user tweeted both pieces of news. The network includes both fake and benign news, with fake news regarded as feature noise induced by malicious user intent ($\epsilon_X$).
>
> To investigate noise dependency patterns associated with the presence of fake news, We hypothesize that the presence of fake news (i.e., feature noise) leads to noisy graph structures and noisy node labels in news networks. Specifically, we assign a semantic topic to each news article as a node label using k-means clustering over BERT embeddings of the article content. For each node in the graph, we compute the Shannon entropy of the semantic topic distribution among its neighboring nodes. We then compare these entropy values between fake and benign news nodes (results shown below).
>
> **Fake news**
> - mean: 1.330333
> - 25%: 1.401647
> - 50%: 1.465137
> - 75%: 1.494437
>
> **Benign news**
> - mean: 1.083667
> - 25%: 1.004271
> - 50%: 1.352713
> - 75%: 1.431281
>
> **Mann-Whitney U test**
> - p-value = 4.186423672941698e-25
> - Mann-Whitney U statistic = 47085.5
>
> Descriptive statistics reveal that fake news nodes generally exhibit higher entropy than benign news nodes, suggesting that benign news tends to connect to semantically similar articles (homophilic), whereas fake news is more frequently connected to semantically dissimilar articles (heterophilic). This observation aligns with common user behavior: people typically share news related to their interests, whereas fake news is often propagated indiscriminately, regardless of topical relevance [2]. Furthermore, a non-parametric statistical test (Mann–Whitney U test) confirms that the difference in entropy values between fake and benign news is statistically significant. These findings suggest that the presence of fake news (i.e., node feature noise) introduces noisy and heterophilic edges into the graph structure. Furthermore, model-based automated news topic prediction often performs poorly due to noise in both features and graph structures, ultimately resulting in incorrect label annotations.
>
> In summary, **these findings empirically support the noisy dependency scenario in real-world scenario where feature noise (i.e., fake news content) can propagate through the graph, generating noisy edges and noisy labels.** This highlights the need for our work that explicitly model and mitigate such noise dependencies in real-world networks.
>
>
> [1] DECOR: Degree-Corrected Social Graph Refinement for Fake News Detection, KDD 2023
>
> [2] Revisiting Fake News Detection: Towards Temporality-aware Evaluation by Leveraging Engagement Earliness, WSDM 2025
>
> ---
>
> ## Response to Q2
>
> In response to the reviewer’s comment, we conduct an experiment where we independently double each of the following: (1) the number of fraudsters (i.e., nodes with noisy features) and (2) the activeness of fraudsters (i.e., the amount of structure noise they introduce) in our real-world DANG generation process.
> As a result, label noise also increases accordingly, in proportion to the amount of generated feature and structure noise.
>
> As shown in the table below, DA-GNN demonstrates competitive performance and, in many cases, outperforms other baselines under these intensified noise conditions.
>
> |        |        |   AirGNN   |    RSGNN   |   STABLE   |   EvenNet  |    NRGNN   |    RTGNN   |   DA-GNN   |
> |--------|--------|:----------:|:----------:|:----------:|:----------:|:----------:|:----------:|:----------:|
> |  Auto  | DANG w/ doubled # frauds | 54.6±1.5 | 53.4±0.7 | 55.4±0.1 | 56.5±0.6 | 55.9±1.5 | 54.3±2.7 | **60.1±0.7** |
> |        | DANG w/ doubled structure noise | 56.9±0.7 | 50.9±0.6 | **58.1±2.0** | 53.6±2.2 | 55.8±1.3 | 56.78±0.8 | 55.6±1.0 |
> | Garden | DANG w/ doubled # fraudsters  | 57.1±1.3 | 65.0±0.5 | 69.8±2.3 | 69.3±1.8 | 70.8±0.7 | 70.6±0.9 | **71.9±0.6** |
> |        | DANG w/ doubled structure noise | 69.9±2.8 | 69.4±1.3 | 72.0±0.5 | 72.4±0.9 | 71.0±2.4 | **75.3±0.4** | 74.4±0.2 |
>
> ---
>
> ## Response to Q3
>
> We conducted sensitivity analyses on our tuned hyperparameters, $\gamma$, $k$, $\theta$, $\lambda_1$, and $\lambda_2$ in Section 5.4 and Appendix F.6. And we found that **DA-GNN is generally not sensitive to most of them**. Hence, we suggest that DA-GNN is practical and easy to deploy, as it does not require extensive hyperparameter tuning, even in low-resource settings.
>
> ---
>
> ## Response to Q4
>
> We apologize for the oversight. We will revise Figure 3 by enlarging the font size and adjusting the layout to improve readability. We also appreciate your observation regarding the missing arrow in Figure 3; we will correct this in the updated version.
>
> ---
>
> ## Response to Q5
>
> In response to the reviewer’s concern, we conduct analysis substituting GNN components with GAT. In the below table, we can see that DA-GAT (DA-GNN with GAT) still outperforms other baselines, which demonstrates there is no sensitivity on specific choice of backbone GNNs.
>
> |          |           |  AirGNN  |   RSGNN  |  STABLE  |  EvenNet |   NRGNN  |   RTGNN  |  SG-GSR  |    DA-GCN    |    DA-GAT    |
> |----------|-----------|:--------:|:--------:|:--------:|:--------:|:--------:|:--------:|:--------:|:------------:|:------------:|
> |   Cora   |   Clean   | 85.0±0.2 | 86.2±0.5 | 86.1±0.2 | 86.2±0.0 | 86.2±0.2 | 86.1±0.2 | 85.7±0.1 |   86.2±0.7   | **86.4±0.2** |
> |          | DANG-10\% | 79.7±0.5 | 81.9±0.3 | 82.2±0.7 | 80.7±0.7 | 81.0±0.5 | 81.6±0.5 | 82.7±0.1 |   82.9±0.6   | **83.0±0.2** |
> |          | DANG-30\% | 71.5±0.8 | 71.9±0.5 | 74.3±0.3 | 65.2±1.7 | 73.5±0.8 | 72.1±0.6 | 76.1±0.2 | **78.2±0.3** |   76.4±0.6   |
> |          | DANG-50\% | 56.2±0.8 | 58.1±0.2 | 62.8±2.4 | 47.1±1.8 | 61.9±1.4 | 60.8±0.4 | 64.3±0.5 | **69.7±0.6** |   69.2±0.3   |
> | Citeseer |   Clean   | 71.5±0.2 | 75.8±0.4 | 74.6±0.6 | 76.4±0.5 | 75.0±1.3 | 76.1±0.4 | 75.3±0.3 | **77.3±0.6** |   76.1±0.2   |
> |          | DANG-10\% | 66.2±0.7 | 73.3±0.5 | 71.5±0.3 | 71.1±0.4 | 71.9±0.3 | 73.2±0.2 | 74.2±0.5 | **74.3±0.9** |   73.4±1.2   |
> |          | DANG-30\% | 58.0±0.4 | 63.9±0.5 | 62.5±1.4 | 61.2±0.6 | 62.5±0.7 | 63.5±2.1 | 65.6±1.0 |   65.6±0.6   | **67.5±1.2** |
> |          | DANG-50\% | 50.0±0.6 | 55.3±0.4 | 54.7±1.7 | 47.2±1.1 | 52.6±0.9 | 54.2±1.8 | 54.8±1.8 | **59.0±1.8** |   57.2±0.5   |
>
>
> Furthermore, we compare our method with representative noise-robust GNN approaches on two heterophilic datasets, Cornell and Wisconsin. As shown in the table below, our proposed method, DA-GNN, achieves superior or competitive performance relative to the baselines, demonstrating its broad applicability across diverse settings.
>
> |         | Cornell    | Wisconsin  |
> |---------|------------|------------|
> | AirGNN  | 52.3±5.6 | 61.4±3.3 |
> | RSGNN   | 71.2±3.4 | **77.8±0.9** |
> | STABLE  | 67.6±0.0 | 75.2±1.1 |
> | NRGNN   | 64.0±1.3 | 72.6±4.8 |
> | RTGNN   | 64.4±5.4 | 64.7±0.0 |
> | SG-GSR  | 45.1±5.6 | 59.5±0.9 |
> | DA-GNN  | **72.1±1.3** | 75.2±1.8 |
>
> ---
>
> ## Response to Q6
>
> In response to the reviewer’s comment, we conduct a new analysis in which we significantly increase or decrease the influence of noise dependency. Due to length constraints, we kindly refer the reviewer to our response to Reviewer KCjc’s comment W2 for further details.

---

> > ### Comment · Reviewer_KkeS · 2025-08-04
> >
> > Thank you for the authors’ detailed responses, which have addressed most of my concerns. However, I still have a few remaining questions and suggestions:
> >
> > - The datasets used in both the main paper and in Response to Q1—such as citation networks and news networks—are all relational networks, where edges reflect social interactions or citations. This raises the question of whether the proposed method is applicable to other types of graphs, such as biological graphs (e.g., protein–protein interaction or molecular structure networks), where noise patterns and causal dependencies may differ. I suggest that the authors clarify the scope of applicability of their method, either in the main text or in the appendix, and discuss to what extent their assumptions generalize to non-relational domains.
> >
> > - As raised in Q3, I would appreciate more clarity on whether the proposed method offers any advantage or disadvantage in terms of runtime and memory consumption, particularly on large-scale datasets like ogbn-arxiv. Since DA-GNN involves multiple encoders, decoders, and latent variables, a brief analysis (e.g., complexity estimate or runtime comparison table) would help assess its practicality.
> >
> >  - The results in Tabs.4 and 5 in response to Reviewer KCjc’s comment W2 demonstrate the advantage of DA-GNN in handling dependent noise (as Tab.3 performs overall better than Tab.4). However, why are the results under the "without noise dependency" trial in Tab.2 (in response to Reviewer KCjc’s comment W2) generally better than those in Table 1 of the main paper?
> >
> > - Regarding the question in Q5, could the authors provide a brief explanation or hypothesis as to why the proposed method is not sensitive to the choice of GNN backbones in the encoder/decoder, as well as to the graph's homophily or heterophily?

---

> ### Author Response · Authors · 2025-08-07
> **re: Official Comment by Reviewer KkeS (1/2)**
>
> We appreciate the reviewer's thoughtful discussion and interest in our work. These insights will meaningfully contribute to improving the quality of the paper.
>
> ### Response to Q1
>
> We thank the reviewer for this valuable suggestion. As rightly pointed out, we acknowledge that not all graph domains strictly conform to the noise assumptions in DANG, for instance, non-relational graphs such as protein–protein interaction or molecular structure networks may exhibit different noise patterns and causal dependencies. However, we would like to emphasize that the DANG framework is well-suited to a wide range of graph domains (see Appendix C.1), particularly those where graph learning methods are most commonly applied, such as citation, social networks, e-commerce, and web graphs.
>
> That said, we fully agree with the reviewer’s suggestion to clarify the scope of applicability and discuss the extent to which the assumptions made in DANG generalize to non-relational domains. We will modify the line 154-157 of Section 3 into the following:
>
> > **As-is:**
> >
> > DANG is prevalent across various domains, including social networks, e-commerce, web graphs, citation networks, and biology. Due to space constraints, detailed scenarios are provided in Appendix C.1. While not all noise scenarios perfectly align with DANG, such cases are rare compared to the widespread occurrence of DANG in these domains, dominating graph applications.
>
> > **To-be:**
> >
> > DANG is prevalent across various domains, including social networks, e-commerce, web graphs, citation networks, and biology networks (e.g., cell-cell graph in the domain of single-cell RNA-sequencing). Due to space constraints, detailed scenarios are provided in Appendix C.1. We acknowledge, however, that not all noise scenarios perfectly align with DANG, and certain edge cases exist. For instance, in non-relational domains such as molecular structures or protein–protein interaction networks, the graph structure is inherently determined and thus unaffected by node feature noise. In such cases, the DANG scenario may not apply.
> >
> > That said, we emphasize that these exceptions are relatively rare compared to the broad applicability of DANG across commonly studied graph domains, including social networks, e-commerce, web graphs, citation networks, and cell-cell network in biology.
>
> ---
>
> ### Response to Q2
>
> **Advantage**
>
> Several baseline methods encounter out-of-memory (OOM) errors on large-scale datasets such as ogbn-arxiv primarily due to their reliance on structure learning components. These methods often require iteratively computing an explicit 𝑁×𝑁 similarity matrix or graph refinement operations, which incur high memory overhead, where 𝑁 is the number of nodes.
>
> In contrast, our method avoids this bottleneck by approximating the regularization term $kl(q_{\phi_1}(Z_A|X, A)||p(Z_A))$ efficiently (See Appendix B). This design significantly reduces the computational and memory cost, enabling DA-GNN to scale to large graphs without OOM issues.
>
> **Disadvantage**
>
> As the reviewer rightly pointed out, the use of multiple encoders and decoders introduces additional computational overhead. However, these components are lightweight compared to the structure learning modules used in many baselines.
>
> To quantify these trade-offs, we conduct an inference time analysis on the ogbn-arxiv dataset, comparing only models that do not suffer from OOM issues. The results are as follows:
>
> DA-GNN: 0.0599 sec
> RTGNN: 0.0461 sec
> AirGNN: 0.0346 sec
> EvenNet: 0.0027 sec
>
> While DA-GNN incurs slightly higher inference time than RTGNN and AirGNN, the difference remains within a practically acceptable range, especially considering that DA-GNN significantly outperforms these models in terms of robustness. Moreover, DA-GNN achieves substantial memory efficiency, avoiding OOM issues that affect 7 out of 10 baseline methods.
>
> **Summary**
>
> Although DA-GNN’s encoder-decoder architecture introduces a modest increase in runtime, this overhead is negligible when weighed against its superior robustness. Furthermore, the efficient design of our regularization component contributes to its strong memory performance on large-scale datasets.

---

> ### Author Response · Authors · 2025-08-07
> **re: Official Comment by Reviewer KkeS (2/2)**
>
> ### Response to Q3
>
> Thank you for the valuable comments. In the “without noise dependency” scenario, there are no causal relationships among $X \rightarrow A$, $A \rightarrow Y$, or $X \rightarrow Y$. This implies that the observed graph includes only independent feature noise ($\epsilon \rightarrow X$), independent structure noise ($\epsilon \rightarrow A$), and NO label noise at all. Please note our assumption in DANG that $\epsilon$ is not a cause of $Y$, which reflects real-world situations where mislabeling is more likely to arise from confusing or noisy features rather than arbitrary sources (see lines 147–150 in the main paper).
>
> Since label noise is known to significantly degrade GNN performance, it is expected that the results in Table 2 (in response to Reviewer KCjc’s comment W2) are generally better than those in Table 1 of the main paper.
>
> Furthermore, we evaluate an extreme noise scenario in Figure 5 of the main paper, where node features, graph structures, and node labels all contain independent noise. In this setting as well, DA-GNN consistently outperforms all baselines.
>
> ---
>
> ### Response to Q4
>
> Thanks for the meaningful discussion. We would like to emphasize that the effectiveness of our proposed method does not stem from the use of a powerful encoder or decoder architecture, but rather from its ability to model the causal relationships inherent in the DGP of DANG. Specifically, each encoder is responsible for encoding a specific latent variable, and each decoder for decoding its corresponding observable variable. These components are jointly trained under the supervision of our objective to effectively fulfill their designated roles. In this context, we claim that the performance of DA-GNN is not sensitive to the choice of GNN backbone.
>
> Furthermore, we attribute the strong performance of DA-GNN on heterophily graphs to the regularization applied to $Z_A$. Specifically, to promote accurate inference of $Z_A$, we regularize it to prioritize assortative edges based on $\gamma$-hop subgraph similarity (see lines 211–217 in the main paper for details). Here, $\gamma$ is set as a hyperparameter in {$0, 1$}. It is widely recognized that, in heterophily graphs, nodes with high feature similarity are more likely to share the same label [1]. In this context, 0-hop subgraph similarity corresponds to feature similarity, which facilitates effective inference of the latent graph structure even in heterophilous settings. This design choice contributes to the strong performance of DA-GNN under heterophily.
>
> [1] Graph Neural Networks for Graphs with Heterophily: A Survey

---

> > ### Comment · Reviewer_KkeS · 2025-08-08
> >
> > The reviewer thanks the authors for their timely and clear response. The rebuttal addressed my concerns. Based on this, I have decided to raise my rating.

---

### Official Review · Reviewer_Ar1L · 2025-06-22

**Clarity:** 2
**Significance:** 3
**Originality:** 2
**Rating:** 5
**Confidence:** 3

**Summary:**

Typically when applying GNN models to real world graph data, the fact that we make noisy observations of some underlying "true" data or structure is ignored, and as a result there is a lack of robustness to these architectures when used out of the box. While there has been some prior work to try and handle this robustness, they assume that the node features and the underlying adjacency structure of the graph are independent, which is an implausible assumption in practice. (For instance, in a citation network node features could correspond to the paper, and so the adjacency structure will have some dependence on the similarity of the underlying papers to each other.) The authors of the current paper propose a more realistic data generating process to capture the noise in these networks, and develop a methodology making use of VAEs in order to perform inference on the graph. The authors also additionally introduce two new datasets to help evaluate robust inference methods for on noisily observed graphs.

**Questions:**

Questions:
1. One part of the proposed data generating process which gives me some pause is the fact that the latent (or "true") node labels $Z_Y$ are causes of both $X$ and $Y$ within the DGP, with $X$ then an additional cause of $Y$. In the scenario where $Y$ is actually truly observed - so $Y = Z_Y$ in some sense - how should we think about the proposed DGP then? Can we just collapse $X$ and $Y$ together into a "single $X$" now? Or should $Z_Y$ actually be thought of as some noise, rather than as "latent clean node labels" as discussed in the introduction?
2. As someone who is unfamiliar with the other methods considered within the experiments section, why do a lot of the models OOM on the arXiv dataset? More generally, I am curious about the inference time of each of the methods, and how these compare across the different models on larger graphs than the Cora dataset (which is very small). Do you have any comments on this?
3. There are various points at which additional regularization are added to the models - for instance, regularizing $Z_A$ to incorporate mostly assortative edges, the regularization of $Z_Y$ to satisfy class homophily, and the label smoothing in the decoder. To what extent does the performance of your method depend on these? Do the methods you compare against perform similar regularization steps? I ask this as when in the scenarios without generative noise added, I don't see any a-priori reason why the current method should do better than existing ones - or even just a regular GCN. (I would be curious to see the experimental results also with a regular GCN applied, to just help illustrate why approaches like those proposed by the authors are needed.) I am trying to gauge the added value singularly from the particular generative model and inference approach proposed. Maybe this is something which can be inferred from Appendix F.2 and F.6, and I am just having a difficult time combining these together, in which case I would appreciate some further elaboration.
4. Did you perform experiments with other ways of generating noise? For example, the provided example the inject noisy features into nodes flips nodes with the same probability as the number of set bits within the node vector. This means that e.g if a vector is mostly all ones, then it will become mostly all zeros afterwards, and so the extent to which a node vector can change is not equal across all nodes. What occurs if some fixed percentage of bits is flipped across all of the nodes?

Some general remarks on the presentation of the paper:
- Within Table 1, it appears that the bolded items only occur if the point estimates for the classification accuracy is the greatest, without incorporating the uncertainty under those estimates. It would help if there were some additional highlighting for the items where the classification accuracy has overlaps in terms of their error bars - for example between SG-SGR and DA-GNN on the DANG-10% Citeseer dataset example.

**Ethical Concerns:**

["NO or VERY MINOR ethics concerns only"]

**Final Justification:**

As discussed in my overall review, I find that the overall contributions of the paper to be significant, in that they provide mechanisms for handling robustness within noisy graph labels and signals, which are demonstrated to be better than existing methods through substantial simulation experiments and studies. While not major concerns of mine which would prevent the acceptance of the paper, the authors of the paper have addressed some of my questions as it relates to the presentation of the paper, which will improve the readability of the paper overall to focus on the key messaging, and I appreciate being addressed by the authors.

**Limitations:**

The authors have adequately addressed the limitations of their work.

**Paper Formatting Concerns:**

I have no concerns over the formatting of the paper.

**Quality:**

4

**Strengths And Weaknesses:**

Strengths of the paper:
- While the key idea in the paper is not that deep - in that it proposes a simple but realistic data generating process, and then derives the consequences of that from a modeling perspective - the method proposed is effective, and leads to improvements in performance over existing baselines. This is especially the case in scenarios where there is a relatively high amount of noise.
- The discussion in the paper as it relates to the modeling approach and the experiments are very detailed and extensive, and cover a significant number of topics as it relates to: the training of the model; ensuring the scalability of the model inference; the effects of removing parts of the generative process and the corresponding drop in performance; hyperparameter studies and so on. I appreciate the extent to which the authors spent their time and effort in studying their method and illustrating it's performance and behavior.

Weaknesses of the paper:
- This is perhaps more a criticism of the datasets examined rather than the methodology itself, but with the exception of the arXiv dataset, the datasets examined are relatively small, with ~10K nodes or fewer (this including the two datasets introduced by the authors). It would be interesting to explore the performance of the method on some more larger graph datasets, such as the Reddit dataset (with ~200K nodes and ~100M edges).
- Generally the presentation in the paper is a bit "squashed" - it tries to cover a lot of topics briefly and provide more details in the Appendix. It would help the readability of the paper if some areas had more time covered in the paper itself, and some were delegated to the Appendix entirely - for example the various details on regularization, as while important, makes it difficult to understand all the different parts of the model and how they are tied together. Moreover, it would help if some of the tables/figures were larger, or had larger versions within the Appendix themselves.

---

> ### Author Rebuttal · Authors · 2025-07-31
>
> We sincerely thank the reviewer for their time and thoughtful review of our paper, as well as for recognizing the contributions of our work. Below, we provide our detailed responses to the reviewer’s concerns and questions.
>
> ## Response to W2
>
> We agree that the current version may appear compressed, as it attempts to cover multiple aspects of our framework within the limited space. In response, we have revised the manuscript to improve clarity and readability by restructuring the content as follows:
>
> - We will streamline the main text to focus more clearly on the core components of our method, ensuring that the key ideas are presented with sufficient depth and continuity.
>
> - To reduce fragmentation and improve conceptual clarity, we will move several technical details on regularization—while still important—to the Appendix, and clearly reference them from the main text.
>
> - We will also enlarge key figures and tables in the main paper where space permits, and add larger, high-resolution versions of them in the Appendix to enhance legibility.
>
> We believe these changes significantly improve the readability of our paper and hope they address the reviewer’s concern.
>
> ---
>
> ## Response to Q1
>
> In our model, $Z_Y$​ represents the true latent label, while $Y$ is the observed label. The observed labels are provided in many research benchmark datasets, which are commonly treated as ground truth for supervised training and evaluation. However, our claim is that even these observed labels $Y$ may contain noise and thus do not reflect the true latent labels.
>
> For example, consider an image of a wolf that is visually similar to a husky. A human annotator might mistakenly label it as a husky due to the visual resemblance. In this case, the observed label $Y$ is “husky,” but the true latent label $Z_Y$​ is “wolf.” The similarity in appearance (i.e., the node feature $X$) has caused incorrect labeling. At the same time, the fact that the true latent label is “wolf” explains why the image depicts a wolf. This kind of mechanism implies that $Z_Y$​ causes both $X$ and $Y$ in the data generating process, with $X$ further contributing to the generation of $Y$. Such a causal structure is well-aligned with the literature on instance-dependent label noise [1] and thus conceptually grounded.
>
> However, because benchmark datasets do not provide ground-truth latent labels or explicit annotations of label noise, we design DANG to simulate this realistic noise scenario and study its impact in a controlled setting.
>
> [1] Instance-dependent Label-noise Learning under a Structural Causal Model, NeurIPS 2021
>
> ---
>
> ## Response to W1 and Q2
>
> Several baseline methods encounter out-of-memory (OOM) errors on large-scale datasets such as ogbn-arxiv primarily due to their reliance on structure learning components. These methods often require computing an explicit 𝑁×𝑁 similarity matrix or graph refinement operations, which incur high memory overhead, where 𝑁 is the number of nodes. In contrast, our method avoids this bottleneck by approximating the regularization term $kl(q_{\phi_1}(Z_A|X, A)||p(Z_A))$ efficiently (See Appendix B). This design significantly reduces the computational and memory cost, enabling DA-GNN to scale to large graphs without OOM issues.
> Furthermore, we conduct an inference time analysis on the arXiv dataset using only models that do not incur OOM errors. The results are as follows:
> - DA-GNN: 0.0599 sec
> - RTGNN: 0.0461 sec
> - AirGNN: 0.0346 sec
> - EvenNet: 0.0027 sec
>
> DA-GNN takes approximately 0.06 seconds for a single inference, which is comparable to RTGNN and AirGNN in terms of inference efficiency. Furthermore, we plan to include a larger-scale graph in our experimental setup to further assess the scalability of DA-GNN.
>
> ---
>
> ## Response to Q3
>
> Some prior methods adopt regularization ideas similar to ours, but they typically apply only a subset of them or rely on different assumptions. For example, RSGNN employs a regularization approach similar to ours for $Z_A$, but it heavily depends on the assumption of strong feature smoothness. Furthermore, it does not incorporate other regularizations. The reason our model achieves strong performance even under clean settings is that our regularization components are not tailored solely for noisy scenarios. Regularizing $Z_A$ to favor assortative edges is closely related to graph structure learning, which has been shown to benefit GNN performance even when no noise is present. Similarly, our regularization on $Z_Y$ encourages class homophily, a widely accepted inductive bias in graph learning, and our decoder for $A$ applies label smoothing to prevent overfitting. These components work synergistically to improve generalization and robustness. Therefore, even in clean settings, DA-GNN outperforms baselines that do not fully leverage these forms of inductive bias.
>
> ---
>
> ## Response to Q4
>
> Due to the nature of bag-of-words features, most node features are very sparse, and cases where a node feature is mostly all ones—as mentioned by the reviewer—are extremely rare in real-world scenarios. If we were to flip a fixed percentage of bits based on high-bit-count nodes, this would lead to overly aggressive perturbation for the majority of nodes, which is unrealistic in practical settings. Conversely, if we set the fixed percentage based on low-bit-count nodes, nodes with richer features would receive almost no perturbation. However, since such high-bit-count nodes are rare, we expect the difference between the fixed-percentage approach and our current method to be minimal in practice.
>
> Additionally, in datasets like arXiv that use numerical node features, we inject Gaussian noise with equal magnitude across all nodes. Even under this setting, DA-GNN consistently outperforms baselines, which further supports the robustness of our approach.
>
> ---
>
> ## Response to Q5
>
> We reported standard deviations alongside point estimates, although for readability we boldfaced only the latter. We agree that including results of statistical significance tests could further clarify performance differences and will consider adding them in a future revision. We would also like to highlight that while this concern may arise under low noise rates, at higher noise levels, DA-GNN consistently outperforms baselines even when accounting for variance.

---

> ### Comment · Reviewer_Ar1L · 2025-07-31
>
> Thanks for the detailed response to my comments and questions. I particularly appreciate the clarifying example given in response to Q1, in addition to the discussion in Q4 as to appropriate noise mechanism (given the underlying features within the produced examples). I am glad to hear the changes you plan on making with regards to the presentation (both as brought up by myself and some of the other reviewers). I will keep my score as-is in support for the acceptance of the paper.

---

### Official Review · Reviewer_KCjc · 2025-07-02

**Clarity:** 3
**Significance:** 2
**Originality:** 3
**Rating:** 4
**Confidence:** 4

**Summary:**

The authors study the practically relevant setting of feature, structure, and label noise for GNNs, where the different noise terms are not assumed to be independent. This addresses the shortcoming in prior work that relies on such simplifying assumptions. To alleviate this shortcoming and train a robust model, the authors propose a latent variational model that may correct noise. The authors demonstrate the empirical efficacy using 7 datasets and 10 baselines.

**Questions:**

1. To what extent are the clean features, edges, labels used for/available while training the model, if the authors study a transductive setting?
1. Did the authors study different constructions of the noise model? E.g., make the feature noise dependent on the graph structure?

**Ethical Concerns:**

["NO or VERY MINOR ethics concerns only"]

**Final Justification:**

Adaptive attacks/evaluation is slightly suspicious since the attack is not very strong. Other concerns have been resolved and the paper may have impact/provide benefit to the respective subfield.

**Limitations:**

yes

**Paper Formatting Concerns:**

Eq.1 violates the margin and is generally quite dense. Perhaps introduce line break before each new term.

**Quality:**

2

**Strengths And Weaknesses:**

The authors address the important problem that previous attempts on modelling noisy features, graph structure, and labels relied on unrealistic assumptions (independence). The paper is well structured and easy to follow. The proposed model and its derivation looks largely sound to me but variation inference is not my research focus.

My main criticism is in the evaluation:
1. Although adversarial robustness is not the key focus of the works, given how the authors phrase the problem, the evaluation should include adaptive adversarial evaluations as previous defenses have shown to be non robust if properly evaluated [1, 2].
2. The evaluation of other noise types could be more extensive. Generally, their DANG model chooses a few constants that should be ablated.
3. It is not clear if the evaluation is transductive/inductive. If following an transductive evaluation one has to be careful that the clean features, edges, labels are not being leaked during training. E.g., see [3] or [4]:
>  transductive setting, i.e., test nodes (except for their labels) are available during training. In this case, defenders can simply memorize benign nodes and identify the injected nodes, making it an imperfect setting.

[1] Tramer et al. “On Adaptive Attacks to Adversarial Example Defenses” NeurIPS 2020

[2] Mujkanovic et al. “Are Defenses for Graph Neural Networks Robust?” NeurIPS 2022

[3] Gosch et al. “Adversarial Training for Graph Neural Networks: Pitfalls, Solutions, and New Directions” NeurIPS 2023

[4] Zheng et al. “Graph Robustness Benchmark: Benchmarking the Adversarial Robustness of Graph Machine Learning” NeurIPS 2021

---

> ### Author Rebuttal · Authors · 2025-07-31
>
> We sincerely thank the reviewer for their time and thoughtful review of our paper.
>
> ## Response to W1
>
> In response to the reviewer’s comment, we conduct experiments on adaptive attacks. Following [1], we implement adaptive PGD attacks on NRGNN, RSGNN, RTGNN, SG-GSR, and DA-GNN using a 10% attack budget.
> As shown in the table below, DA-GNN demonstrates comparable or superior adversarial robustness against adaptive attacks compared to the baselines.
>
> |          |    RSGNN   |    NRGNN   |    RTGNN   |     SG-GSR     |     DA-GNN     |
> |----------|:----------:|:----------:|:----------:|:--------------:|:--------------:|
> |   Cora   | 82.68±0.41 | 80.73±0.23 | 81.90±0.32 | **83.23±0.78** |   82.56±0.17   |
> | Citeseer | 74.35±0.72 | 70.75±0.23 | 71.00±0.10 |   73.60±1.11   | **74.59±0.29** |
>
> [1] Are Defenses for Graph Neural Networks Robust?, NeurIPS 2022
>
> ---
>
> ## Response to W2
>
> In the generation process of our synthetic DANG, we have three variables: 1) the overall noise rate, 2) the amount of noise dependency ($X \rightarrow A, X\rightarrow Y, A \rightarrow Y$), and 3) the amount of independent structure noise ($\epsilon \rightarrow A$).
> - For the first variable, our experiments already addressed it by varying the noise rate from 0 to 50.
> - For the second variable, we conduct an additional analysis by substantially increasing or decreasing the degree of noise dependency. Specifically, we increase the number of structure noise edges caused by feature noise by approximately 4×, and similarly amplify the amount of label noise induced by both feature and structure noise by 4×. We also evaluate a setting where noise dependencies are completely removed—this corresponds to a scenario with independent feature and structure noise. As shown in Table 1 and Table 2, DA-GNN consistently outperforms all baselines on the strong presence of noise dependency, and shows competitive performance on the weak presence of noise dependency.
>
> **Table 1. Results on DANG with increased noise dependency**
>
> |          |           |  AirGNN  |   RSGNN  |  STABLE  |  EvenNet |   NRGNN  |   RTGNN  |  SG-GSR  |  DA-GNN  |
> |:--------:|-----------|:--------:|:--------:|:--------:|:--------:|:--------:|:--------:|:--------:|:--------:|
> |   Cora   | DANG-10\% | 78.3±0.3 | 79.0±0.1 | 79.5±0.4 | 76.9±1.2 | 78.6±0.5 | 78.8±0.5 | 78.5±0.2 | **79.8±0.2** |
> |          | DANG-30\% | 57.6±0.5 | **67.9±0.6** | 65.2±1.4 | 55.8±1.3 | 63.8±0.9 | 66.1±0.6 | 56.9±0.7 | 67.0±0.3 |
> |          | DANG-50\% | 40.1±0.5 | 49.4±0.9 | 45.7±1.2 | 40.5±1.0 | 47.5±0.5 | 48.1±0.8 | 40.1±1.2 | **51.6±0.9** |
> | Citeseer | DANG-10\% | 65.7±1.1 | **72.9±0.4** | 68.4±0.7 | 68.8±0.6 | 69.7±1.0 | 69.8±0.0 | 70.3±0.5 | 72.7±0.3 |
> |          | DANG-30\% | 57.2±0.9 | 63.3±0.6 | 57.2±0.1 | 57.2±0.5 | 59.6±0.7 | 60.1±0.7 | 62.0±0.9 | **64.9±0.6** |
> |          | DANG-50\% | 39.8±0.7 | 49.4±1.0 | 41.3±1.8 | 42.2±0.5 | 42.9±0.6 | 43.7±0.7 | 46.1±1.3 | **51.4±0.2** |
>
> **Table 2. Results on DANG without noise dependency**
> |         |           |  AirGNN  |     RSGNN    |  STABLE  |  EvenNet |   NRGNN  |     RTGNN    |    SG-GSR    |    DA-GNN    |
> |:--------:|-----------|:--------:|:------------:|:--------:|:--------:|:--------:|:------------:|:------------:|:------------:|
> |   Cora   | DANG-10\% | 83.1±0.4 |   83.9±0.6   | 83.9±0.4 | 84.0±0.5 | 83.8±0.2 | **84.8±0.2** | **84.8±0.1** | **84.8±0.1** |
> |          | DANG-30\% | 77.7±0.5 |   79.9±0.4   | 76.8±0.6 | 74.8±0.6 | 77.8±0.8 |   80.0±0.1   | **80.1±0.1** | **80.1±0.3** |
> |          | DANG-50\% | 66.9±2.6 |   72.7±0.6   | 70.3±1.8 | 61.3±3.4 | 69.1±0.8 |   73.2±0.6   |   72.0±0.2   | **75.4±0.0** |
> | Citeseer | DANG-10\% | 68.8±0.3 | **75.9±0.9** | 71.9±0.6 | 74.0±0.3 | 74.1±0.7 |   74.4±0.4   |   75.5±0.5   | **75.9±0.4** |
> |          | DANG-30\% | 63.6±0.2 |   70.7±0.5   | 67.3±0.2 | 67.9±0.3 | 69.7±0.3 |   69.4±0.6   | **72.0±0.4** |   71.5±0.3   |
> |          | DANG-50\% | 59.7±0.7 |   64.6±0.7   | 59.3±0.6 | 61.3±0.6 | 63.4±0.4 |   64.6±0.2   | **66.4±0.3** |   64.7±0.6   |
>
>
> - For the third variable, we perform an additional analysis by doubling the amount of independent structure noise. We also evaluate the case where no independent structure noise is present. As shown in Table 3 and Table 4, DA-GNN consistently outperforms all baselines across both settings.
>
> **Table 3. Results on DANG without independent structure noise**
>
> |          |           |  AirGNN  |   RSGNN  |  STABLE  |  EvenNet |   NRGNN  |   RTGNN  |  SG-GSR  |    DA-GNN    |
> |:--------:|-----------|:--------:|:--------:|:--------:|:--------:|:--------:|:--------:|:--------:|:------------:|
> |   Cora   | DANG-10\% | 81.1±0.5 | 81.1±0.6 | 83.1±0.7 | 81.4±0.3 | 82.1±0.3 | 82.6±0.2 | 82.5±0.1 | **83.9±0.3** |
> |          | DANG-30\% | 73.9±1.7 | 73.6±0.3 | 76.9±0.3 | 69.7±0.7 | 76.3±0.4 | 74.8±0.9 | 77.7±0.3 | **79.6±0.6** |
> |          | DANG-50\% | 64.6±2.3 | 60.3±1.3 | 66.4±0.6 | 51.6±0.5 | 64.4±1.0 | 62.8±0.6 | 69.5±1.0 | **72.1±0.4** |
> | Citeseer | DANG-10\% | 68.3±0.6 | 71.8±0.7 | 72.4±0.9 | 72.8±0.1 | 73.1±0.3 | 73.7±0.2 | 74.5±0.4 | **74.7±0.1** |
> |          | DANG-30\% | 58.5±0.5 | 63.5±0.9 | 64.6±0.2 | 63.1±0.4 | 64.3±1.4 | 64.8±0.9 | 66.1±0.6 | **66.4±0.6** |
> |          | DANG-50\% | 54.3±0.2 | 55.9±0.3 | 58.1±1.0 | 51.2±2.1 | 56.7±0.2 | 56.6±0.9 | 59.3±0.6 | **60.3±1.2** |
>
>
> **Table 4. Results on DANG with increased independent structure noise**
> |          |           |  AirGNN  |     RSGNN    |  STABLE  |  EvenNet |   NRGNN  |   RTGNN  |  SG-GSR  |    DA-GNN    |
> |:--------:|-----------|:--------:|:------------:|:--------:|:--------:|:--------:|:--------:|:--------:|:------------:|
> |   Cora   | DANG-10\% | 78.9±0.7 |   81.6±0.3   | 80.9±0.5 | 78.9±0.3 | 79.9±0.4 | 81.8±0.3 | 81.4±0.2 | **82.5±0.2** |
> |          | DANG-30\% | 66.1±1.8 |   70.6±0.9   | 72.0±0.8 | 61.0±0.9 | 72.2±0.6 | 70.1±0.6 | 72.4±0.2 | **75.4±0.4** |
> |          | DANG-50\% | 47.3±0.5 |   56.7±0.2   | 57.9±1.6 | 42.1±2.1 | 58.8±0.6 | 55.6±0.5 | 61.2±1.5 | **65.4±0.6** |
> | Citeseer | DANG-10\% | 66.5±0.4 | **74.0±0.3** | 70.8±0.4 | 70.3±0.8 | 71.9±0.1 | 72.6±0.3 | 72.6±0.4 |   73.6±0.2   |
> |          | DANG-30\% | 58.0±0.2 |   63.8±0.6   | 62.3±1.8 | 60.1±0.3 | 61.6±0.9 | 63.2±0.5 | 63.7±0.8 | **65.1±0.5** |
> |          | DANG-50\% | 49.4±0.6 |   55.1±0.3   | 51.7±1.7 | 45.9±0.7 | 50.8±0.8 | 52.0±1.1 | 54.2±0.2 | **55.4±1.2** |
>
>
> These results demonstrate that DA-GNN consistently outperforms other baselines under varying degrees of DANG, highlighting its practical applicability across diverse real-world noise conditions.
>
> ---
>
> ## Response to W3
>
> We followed **the transductive setting with training-time noise** in features, edges, and labels. There are no leaked clean features, edges, and labels in our setting because our setup fundamentally differs from **the transductive setting with test-time attack**, which is mentioned in [2] and [3]. Specifically, [2] claims that a model can achieve perfect robustness by memorizing the testing nodes and their clean structures, which are leaked during training.
>
> However, our setting assumes that the given graph already incorporates the noise before any training begins. The model never has access to the clean graph at any point. Hence, memorizing the noisy graph is not advantageous in achieving robustness. Instead, our model must learn to identify and handle noise patterns rather than memorize clean structures.
>
> [2] Gosch et al. “Adversarial Training for Graph Neural Networks: Pitfalls, Solutions, and New Directions” NeurIPS 2023
>
> [3] Zheng et al. “Graph Robustness Benchmark: Benchmarking the Adversarial Robustness of Graph Machine Learning” NeurIPS 2021
>
> ---
>
> ## Response to Q1
>
>
> Clean features, edges, and labels are NOT available at all during training. Our model is trained entirely on the noisy graph - we add 10%, 30%, or 50% noise to features, structure, and labels before training begins, and the model never sees the clean version. This ensures no information leakage and forces the model to learn genuine denoising capabilities.
>
> ---
>
> ## Response to Q2
>
> We have considered the scenario where structural noise in the graph influences the node features, as the reviewer suggested, and we discuss this in Appendix C.2. Exploring more diverse constructions of noise models—including those in which feature noise is explicitly dependent on the graph structure—remains an interesting direction for future work.

---

> > ### Comment · Reviewer_KCjc · 2025-08-05
> >
> > I thank the authors for their response and the extensive set of new experimental results! Some follow-up questions:
> >
> > W1) To clarify: you are now running Meta-PGD and back propagate through the training for all models? Or what are you doing specifically?
> >
> > W2) Does the model during training only see a single randomly perturbed version? I.e., are the inputs to Algorithm 1 the outputs of Algorithm 2? If so, it would be great to follow the notation in Algorithm 2 and superscript by "noisy".

---

> > > ### Author Response · Authors · 2025-08-06
> > >
> > > We thank the reviewer for the prompt response and thoughtful follow-up questions.
> > >
> > >
> > > **W1**
> > >
> > > To implement the adaptive attack, we adopt Aux-Attack, an efficient variant of Meta-PGD proposed in [1]. Specifically, we first train each target model using 10% of the training nodes and select the best-performing model based on optimal hyperparameters. We then apply the adaptive attack by perturbing the graph structure using the gradient of the loss computed on the target model itself.
> > >
> > > [1] “Are Defenses for Graph Neural Networks Robust?”
> > >
> > >
> > > **W2**
> > >
> > > The reviewer’s understanding is correct. The output of Algorithm 2 is a noisy graph generated under the DANG framework, which is then used as input to the DA-GNN training algorithm described in Algorithm 1. Hence, during training, the model observes only a single randomly perturbed version of the graph, and no clean features, edges, or labels are leaked.
> > >
> > > We fully agree with the suggestion to follow the notation in Algorithm 2 and to use a “noisy” superscript for clarity. We will incorporate this change in the next version of the paper. We appreciate the valuable feedback.

---

> > > > ### Comment · Reviewer_KCjc · 2025-08-06
> > > >
> > > > I thank for the clarifications!
> > > >
> > > > W1: I appreciate the effort for the authors to conduce these attacks! I am not into all the details of all the models but from what I understand, handling non-differentiable components can be tricky and requires care. Having said this, I am a bit surprised that the attack is weaker at 10% than DANG. It has been a recurring theme in literature that random noise (on the graph structure) is usually rather weak. I kindly ask the authors to review everything and potentially fix any errors.
> > > >
> > > > Even though I am not fully convinced regarding the results with adaptive attacks (e.g., loss curves over attack steps would be helpful), I adjusted my rating, reflecting the clarifications and extra results that strengthen the paper.

---

> > > > > ### Author Response · Authors · 2025-08-07
> > > > >
> > > > > We appreciate the reviewer for acknowledging our efforts in the rebuttal. Regarding the point that the adaptive attack appears weaker than DANG at 10% noise, we would like to clarify that DANG inherently introduces noise across all components, i.e., features, structure, and labels, due to its design. Specifically, DANG assumes that noise in node features may create a chain of noise dependencies that propagate to the graph structure and node labels.
> > > > >
> > > > > Moreover, it is not immediately evident that the adaptive attack is weaker than DANG. As shown in the table below, the adaptive attack achieves performance comparable to that of DANG, demonstrating competitive attack effectiveness. This highlights the strength of the adaptive attack, as it only perturbs the graph structure while still approaching the effectiveness of DANG, which corrupts all components. Therefore, we argue that the performance under the adaptive attack is reasonable.
> > > > >
> > > > > | Dataset  |    Attack    |    RSGNN   |   NRGNN  |   RTGNN  |  SG-GSR  |  DA-GNN  |
> > > > > |----------|:------------:|:----------:|:--------:|:--------:|:--------:|:--------:|
> > > > > |   Cora   | adaptive 10% |  **82.7±0.4**  | 80.7±0.2 | **81.9±0.3** | **83.2±0.8** | 82.6±0.2 |
> > > > > |          |   DANG 10%   |  81.9±0.3  | **81.0±0.5** | 81.8±0.3 | 82.7±0.1 | **82.9±0.6** |
> > > > > | Citeseer | adaptive 10% | **74.4±0.7** | 70.8±0.2 | 71.0±0.1 | 73.6±1.1 | **74.6±0.3** |
> > > > > |          |   DANG 10%   |  73.3±0.5  | **71.9±0.3** | **73.2±0.2** | **74.2±0.5** | 74.3±0.9 |

---

### Official Review · Reviewer_oKo2 · 2025-07-03

**Clarity:** 3
**Significance:** 3
**Originality:** 3
**Rating:** 5
**Confidence:** 4

**Summary:**

This paper proposes dependency-aware noise (DANG) on graphs to overcome the unrealistic assumption that noise in node features is independent of the graph structure and node labels, which creates a chain of noise dependencies that propagates to the graph structure and node labels. Based on this, the authors propose a novel robust GNN, DA-GNN, which captures the causal relationships among variables in the data generating process  of DANG using variational inference. Extensive experiments demonstrate that DA-GNN consistently outperforms existing baselines across various noise scenarios. It was an interesting and more realistic attempt.

**Questions:**

See weaknesses.

**Ethical Concerns:**

["NO or VERY MINOR ethics concerns only"]

**Limitations:**

yes

**Paper Formatting Concerns:**

No formatting issues.

**Quality:**

3

**Strengths And Weaknesses:**

Strengths:
(1) This paper examines the gap between real-world scenarios and the overly simplistic noise assumptions underlying previous robust GNN research, which constrain their practicality.
(2) This paper is well thought out, starting from the problem, to designing the noise dependency chain, and finally proposing a correlation graph neural network.
(3) The article is well laid out with a clear focus and non-focus. Experiments are more adequate to validate the proposed model.

Weaknesses:
(1) Paragraphs 1, 2, 3, and 4 of the introduction are all describing unrealistic assumptions by examples and are overly redundant, please add some description of the relevant literature. Example description redundancy: more theory, literature, empirical description, less example description.
(2) The generalizability and applicability of DA-GNN is mentioned in the introduction, please cite some references appropriately, such as “Unifying Homophily and Heterophily for Spectral Graph Neural Networks via Triple Filter Ensembles”.
(3) The Formulation part of section 3 has word problems, e.g. “Y the observed node labels”. Please check the whole text carefully, if there are more similar problems in the revised version, no acceptance advice will be given.
(4) Please explain how DANG works and its role in the following description (a) and Figure 3: (a) "We propose a novel robust GNN, DA-GNN,which captures the causal relationships among variables in the data generating process (DGP) of DANG using variational inference." In addition, how are the noisy dependency chain of DANG generated and what is their association with the subgraph on the left side of Figure 3.

---

> ### Author Rebuttal · Authors · 2025-07-25
>
> We sincerely thank the reviewer for their time and thoughtful review of our paper, as well as for recognizing the contributions of our work.
>
> ## Response to W1
>
> Our intention with the examples in Paragraphs 1–4 was to intuitively illustrate the limitations of existing assumptions and make the introduction accessible to readers from diverse backgrounds. However, we agree that the current version may place too much emphasis on illustrative examples at the expense of theoretical grounding or empirical context.
>
> In the revised version, we will streamline redundant descriptions and expand the discussion of relevant literature and empirical motivations to better position our work within the broader research landscape. Specifically, we plan to incorporate the empirical findings referenced in our response to Reviewer KkeS, Question #1.
>
> We appreciate your suggestion, which has helped us improve the clarity and balance of the introduction.
>
> ---
>
> ## Response to W2
>
> Thank you for pointing this out. We agree that grounding the discussion of generalizability and applicability with appropriate references is important. To ensure that we fully understand your suggestion and incorporate the most relevant citations, we would greatly appreciate it if you could provide a bit more context on the types of works or specific aspects you believe should be highlighted—particularly in relation to the cited work “Unifying Homophily and Heterophily for Spectral Graph Neural Networks via Triple Filter Ensembles.”
>
> ---
>
> ## Response to W3
>
> We sincerely apologize for the incomplete phrase. The line the reviewer pointed out should be modified to: "In Fig. 2(b), 𝑋 denotes the node features (potentially noisy), 𝑌 denotes the observed node labels (possibly noisy), 𝐴 denotes the observed edges (which may contain noise), and 𝜖 denotes the environment variable that causes the noise."
>
> In addition to this line, we carefully reviewed and corrected the rest of the text in Section 3. We will also conduct a thorough proofreading of the entire manuscript to ensure clarity, completeness, and grammatical correctness throughout. We greatly appreciate your attention to detail, which has helped us improve the overall quality and readability of the paper.
>
> ---
>
> ## Response to W4
>
> **About description (a).**
> The description (a) means that DA-GNN is explicitly designed to model this dependency structure by introducing latent variables $Z_Y$, $Z_A$, and $\epsilon$, and optimizing the evidence lower bound (ELBO) of the full joint distribution 𝑃(𝑋,𝐴,𝑌).
> Specifically, DA-GNN learns the encoder parameters for the latent variables, $\phi=${  $\phi_1, \phi_2, \phi_3$ }, and the decoder parameters for the observable variables, $\theta=$ { $\theta_1, \theta_2, \theta_3$ }, to maximize the ELBO.
> Through this training process, the encoders and decoders are jointly optimized to model how the observed noisy graph is generated from both latent and observable variables, and to capture the causal relationships that induce noise dependencies. As a result, DA-GNN promotes accurate inference of the latent clean node labels $Z_Y$ and latent clean graph structure $Z_A$, enabling effective performance on node classification and link prediction tasks even in the presence of complex noise dependencies.
>
> **About Figure 3.**
> On the left side of Figure 3, we illustrate both the latent clean graph and the observed graph affected by DANG. Importantly, this is not just a subgraph, but a depiction of an observed graph that includes unlabeled nodes, noisy node features, noisy edges, and noisy node labels.
>
> In the latent clean graph, green and blue nodes each represent different classes. Nodes within the same class (e.g., all green or all blue) have similar features and tend to form edges with one another—this reflects a typical property of clean real-world graphs, where class-homophily governs both feature similarity and structural connectivity.
>
> DANG introduces noise into this clean graph by assuming that a latent variable 𝜖 perturbs some node features. In the observed graph, two nodes exhibit noisy features (shown as red blocks), corrupted due to 𝜖. Because these noisy features distort the similarity between nodes, noisy edges (shown as red solid lines) are formed between nodes of different classes (e.g., green and blue), as they now appear deceptively similar.
>
> Furthermore, this feature and structure noise can lead to label noise, where some nodes are incorrectly labeled—represented by circles with a minus sign. This cascading chain of noise dependencies lies at the core of the DANG formulation. It is elaborated in Section 3 (theoretical formulation), Appendix C.1 (conceptual examples), and Appendix E.2 (empirical implementation details).
>
> We hope this response sufficiently addresses the reviewer’s concerns and clarifies the underlying motivation and formulation.

---

> > ### Comment · Reviewer_oKo2 · 2025-08-06
> >
> > I carefully read the author's rebuttal, and I believe it basically answered my questions, so I will maintain my original score.

---

### Comment · Area_Chair_xK6M · 2025-08-05

Hi Reviewers,

If you have not checked the authors' rebuttal, please check and reply. If you do not agree with other reviewers comments, please leave a comment.

Thank you!

AC

---

### Note · Authors · 2025-08-14

Dear Reviewers, AC, and SAC,



We hope the following brief summary of our rebuttal interactions will be helpful for the review process.

We sincerely thank all reviewers for their valuable suggestions and timely responses. Most concerns have been addressed during the rebuttal phase, and the paper has been further strengthened, receiving **positive feedback from the majority of reviewers (two accepts and one borderline accept)**. In particular:

- **Writing improvement:** We have proposed clear revision plans to improve the clarity and readability of the manuscript.

- **Empirical justification of DANG:** We conducted data analyses on news networks, providing empirical evidence for the real-world existence of DANG.

- **Further evaluation of DA-GNN:** We demonstrated the robustness of DA-GNN across diverse scenarios, including:

  - Varying the degree of noise dependency and independent noise in synthetic DANG,

  - Adjusting the number and activity level of fraudsters in real-world DANG,

  - Evaluation on heterophilous graphs,

  - Evaluation under adaptive attacks.

The only remaining question is reviewer KCjc’s observation that the adaptive attack with 10% perturbation appears competitive to DANG with 10% noise, which might suggest a weakness of the implemented adaptive attacks. However, we attribute this to the adaptive attack perturbing only the graph structure, whereas DANG perturbs all graph components. This underscores the adaptive attack’s strength in achieving near-DANG performance while modifying only the structure, making the observed results reasonable.

We are grateful for the reviewers’ recognition of our work’s novelty, effectiveness, and thorough evaluation. We will carefully incorporate them into the revised version of our paper.

Best regards,

Authors

---

### Decision · Program_Chairs · 2025-09-17

**Decision:**

Accept (poster)

**Comment:**

This paper proposes a realistic noise model (DANG) and a corresponding method (DA-GNN) that uses variational inference to capture noise dependencies in features, structure, and labels. The reviewers agree that the problem is important, the method is technically sound, and the experiments are thorough. Concerns about clarity, novelty, and adaptive attack evaluation were raised, but the authors responded with new analyses, empirical evidence, and planned revisions. After the rebuttal, all reviewers were satisfied, with two clear accepts and two borderline accepts raised to support acceptance.